# CAR-Flow: Condition-Aware Reparameterization Aligns Source and Target for Better Flow Matching

Chen Chen    Pengsheng Guo[*]   Liangchen Song    Jiasen Lu    Rui Qian    Xinze Wang

Tsu-Jui Fu[*]        Wei Liu[*]        Yinfei Yang        Alex Schwing

Apple Inc.

## Abstract

Conditional generative modeling aims to learn a conditional data distribution from samples containing data-condition pairs. For this, diffusion and flow-based methods have attained compelling results. These methods use a learned (flow) model to transport an initial standard Gaussian noise that ignores the condition to the conditional data distribution. The model is hence required to learn both mass transport *and* conditional injection. To ease the demand on the model, we propose *Condition-Aware Reparameterization for Flow Matching* (CAR-Flow) – a lightweight, learned *shift* that conditions the source, the target, or both distributions. By relocating these distributions, CAR-Flow shortens the probability path the model must learn, leading to faster training in practice. On low-dimensional synthetic data, we visualize and quantify the effects of CAR-Flow. On higher-dimensional natural image data (ImageNet-256), equipping SiT-XL/2 with CAR-Flow reduces FID from 2.07 to 1.68, while introducing less than $0.6\%$ additional parameters.

## 1   Introduction

Conditional generative models enable to draw samples conditioned on an external variable—for example, a class label, a text caption, or a semantic mask. They are a key technology that has advanced significantly in the last decade, from variational auto-encoders (VAEs) [Kingma and Welling, 2014] and generative adversarial nets [Goodfellow et al., 2014] to diffusion [Ho et al., 2020, Song et al., 2021a,b, Dhariwal and Nichol, 2021, Peebles and Xie, 2023, Rombach et al., 2022, Nichol and Dhariwal, 2021] and flow-matching [Ma et al., 2024, Liu et al., 2023, Lipman et al., 2023, Albergo and Vanden-Eijnden, 2023, Albergo et al., 2023].

State-of-the-art diffusion and flow-matching frameworks accomplish conditional generation by using a trained deep net to trace a probability path that progressively transforms samples from a simple *source* distribution into the rich, condition-dependent *target* distribution. A popular choice for the source distribution is a single, condition-agnostic standard Gaussian. Consequently, the conditioning signal enters only through the network itself: in flow matching, for instance, the model predicts a velocity field where the condition is commonly incorporated via embeddings or adaptive normalization layers. Although this strategy has enabled impressive results, this design forces the network to shoulder two tasks simultaneously—(i) transporting probability mass to the correct region of the data manifold, and (ii) encoding the semantic meaning of the condition. Because different conditions often occupy distant parts of that manifold, the dual burden stretches the learned trajectory, slows convergence, and can impair sample quality and diversity.

---

[*]Work done while at Apple.

39th Conference on Neural Information Processing Systems (NeurIPS 2025).

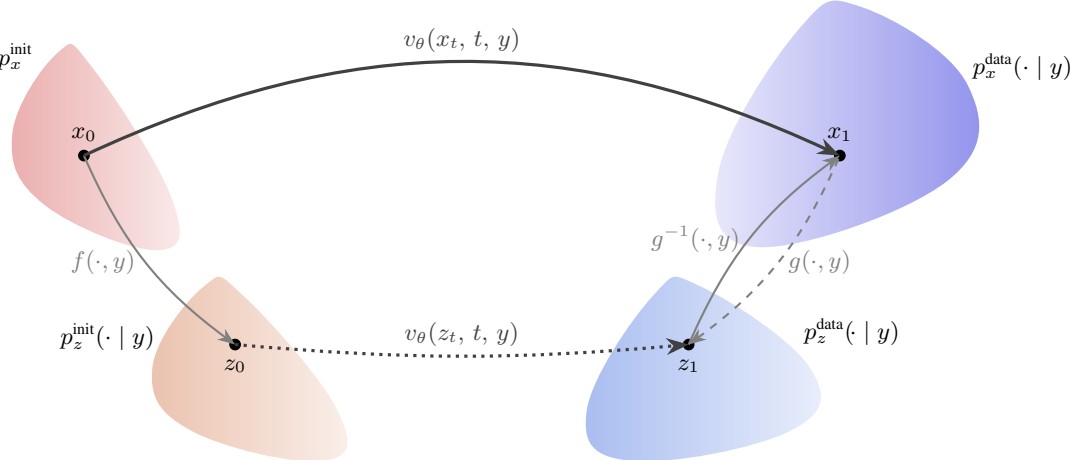

Figure 1: **Condition-Aware Reparameterization for Flow Matching (CAR-Flow).** Illustration of the push-forward under standard conditional flow matching (direct mapping $x_0 \to x_1$) versus our Condition-Aware Reparameterization (CAR-Flow) chain ($x_0 \to z_0 \to z_1 \to x_1$). In the standard setting, a condition-agnostic prior sample (red) is carried by the network's velocity field directly to each condition-dependent data manifold (blue), forcing it to juggle long-range transport and semantic injection at once. CAR-Flow, in contrast, employs lightweight *source distribution map* $f(\cdot, y)$ and *target distribution map* $g(\cdot, y)$ to align the source and target distributions to relieve the network of unnecessary transport. During sampling, $x_1$ is obtained via the (approximate) inverse map $g^{-1}(\cdot, y)$.

In this paper, we alleviate this burden via a condition-dependent mapping of the source and target distributions. First, we endow the source distribution with condition-awareness through a lightweight map, i.e., the *source distribution map*. Hence, every condition has its own source distribution. The same idea can be mirrored to the target distribution using another lightweight map, i.e., the *target distribution map*, that we require to be approximately invertible to keep sampling tractable. The resulting push-forward chain is illustrated in Figure 1. Intuitively, the target distribution map and its approximated inverse correspond to the encoder and decoder, commonly used in latent diffusion pipelines [Rombach et al., 2022]. Differently, here, the target distribution map is explicitly conditional, aligning our formulation with recent latent-space conditioning approaches that embed semantic information directly into the latent representation [Wu et al., 2024, Leng et al., 2025, Yao et al., 2025].

Although fully general maps provide maximal flexibility, this freedom conceals a critical failure mode: the flow-matching objective admits *zero-cost* solutions. The predicted data distribution then collapses to a single mode. A comparable drop in generation quality has been observed in recent work when a variational auto-encoder (VAE) is naively fine-tuned end-to-end with a diffusion model [Leng et al., 2025]. We formalize these collapse routes and later verify them experimentally.

To preclude this trivial solution, we impose the simplest effective constraint—**shift-only conditioning**. Concretely, we reparameterize the maps to only translate. Relocating the start and/or end points while leaving scales untouched removes all trivial solutions, and still shortens the residual probability path. Note that volume-preserving maps are also possible. We refer to the resulting framework as **condition-aware reparameterization for flow matching (CAR-Flow)**.

CAR-Flow can be applied to the source distribution, the target distribution, or both. Each variant shapes the learned path differently, and we find the joint version to perform best in practice. For both low-dimensional synthetic benchmarks and high-dimensional natural image (ImageNet-256) data, CAR-Flow consistently improves performance: augmenting the strong SiT-XL/2 baseline [Ma et al., 2024] with CAR-Flow reduces FID from 2.07 to 1.68 while adding fewer than $0.6\%$ additional parameters.

Our contributions are as follows:

1. We introduce CAR-Flow, a simple yet powerful shift-only mapping that aligns the source, the target, or both distributions with the conditioning variable. CAR-Flow relieves the velocity network of unnecessary transport while adding negligible computational overhead.

2. We provide a theoretical analysis that uncovers *zero-cost* solutions present under unrestricted reparameterization and prove that they render the velocity collapse, thereby explaining the empirical failures of naïve end-to-end VAE-diffusion training.

3. We validate CAR-Flow on low-dimensional synthetic data and the high-dimensional ImageNet-256 benchmark, consistently outperforming the standard rectified flow baseline.

## 2 Background and Preliminaries

We start with a brief review of the standard flow-matching formulation.

### 2.1 Conditional Generation and Probability Paths

Conditional generation seeks to sample from a conditional data distribution $x_1 \sim p_x^{\text{data}}(\cdot \mid y)$, where $y$ is a conditioning variable, e.g., a class label or a text prompt. Diffusion and flow-based models tackle this problem by simulating a differential equation that traces a probability-density path $p_t(x_1 \mid x_0, y)$, gradually transporting samples from a simple source distribution $x_0 \sim p_x^{\text{init}}$ into the target conditional distribution. A standard choice for the source distribution is the isotropic Gaussian $p_x^{\text{init}} = \mathcal{N}(0, I_d)$.

The stochastic trajectory $(X_t)_{0 \leq t \leq 1}$ follows the SDE

$$\mathrm{d}X_t = \left[ u_t(X_t, y) + \frac{\sigma_t^2}{2} \nabla \log p_t(X_t \mid y) \right] \mathrm{d}t + \sigma_t \, \mathrm{d}W_t, \tag{1}$$

where $u_t$ is the drift field, $\sigma_t$ is the diffusion coefficient, and $W_t$ is a standard Wiener process. The term $\nabla \log p_t(X_t \mid y)$ denotes the score function, which can be written in terms of the drift field $u_t$ [Ma et al., 2024]. In the deterministic limit $\sigma_t = 0$, this SDE reduces to the ODE $\mathrm{d}X_t = u_t(X_t, y) \, \mathrm{d}t$.

### 2.2 Gaussian Probability Paths

A convenient instantiation is the *Gaussian path*. Let $\alpha_t$ and $\beta_t$ be two continuously differentiable, monotonic noise scheduling functions satisfying the boundary conditions $\alpha_0 = \beta_1 = 0$ and $\alpha_1 = \beta_0 = 1$. At time $t$, the conditional distribution is

$$p_t(\cdot \mid x_1, y) = \mathcal{N}\big(\alpha_t x_1, \beta_t^2 I_d \mid y\big), \tag{2}$$

whose endpoints are $p_0(\cdot \mid x_1, y) = \mathcal{N}(0, I_d)$ and $p_1(\cdot \mid x_1, y) = \delta_{x_1 \mid y}$. Here $\delta$ denotes the Dirac delta "distribution". Along this path, the state evolves by the interpolant $x_t = \beta_t x_0 + \alpha_t x_1$, with velocity field

$$u_t = \dot{\beta}_t x_0 + \dot{\alpha}_t x_1, \tag{3}$$

where overdots denote time derivatives.

### 2.3 Conditional Flow Matching and Sampling

Conditional flow matching trains a neural velocity field $v_\theta(x, t, y)$ to approximate the true velocity $u_t$ specified in Eq. (3). The commonly employed objective is

$$\mathcal{L}(\theta) = \mathbb{E}_{\substack{y \sim p_Y, \, t \sim p_T \\ x_0 \sim p_x^{\text{init}}, \, x_1 \sim p_x^{\text{data}}}} \left\| v_\theta\big(\beta_t x_0 + \alpha_t x_1, t, y\big) - \big(\dot{\beta}_t x_0 + \dot{\alpha}_t x_1\big) \right\|^2, \tag{4}$$

where $p_Y$ is the marginal distribution over conditions $y$, and $p_T$ is a time density on $[0, 1]$.

After training, one draws $x_1 \sim p_x^{\text{data}}$ by numerically integrating Eq. (1) from $t = 0$ to $t = 1$ with $\hat{u}_t = v_\theta(x_t, t, y)$. A solver-agnostic procedure is outlined in Algorithm 1.

**Algorithm 1** Sampling via conditional flow matching (standard Gaussian source)

---

**Require:** trained network $v_\theta$; diffusion schedule $\{\sigma_t\}_{t\in[0,1]}$; number of steps $N$

1: Sample $y \sim p_Y$
2: Sample $x_0 \sim \mathcal{N}(0, I_d)$
3: $\Delta t \leftarrow 1/N; \quad x \leftarrow x_0$
4: **for** $k = 0$ **to** $N - 1$ **do**
5:     $t \leftarrow k\,\Delta t$
6:     $u \leftarrow v_\theta(x, t, y)$                                   */* drift */
7:     Integrate SDE in Eq. (1) over $[t, t + \Delta t]$ with $(u, \sigma_t)$      */* e.g., Euler–Maruyama */
8: **end for**
9: **return** $x$                              */* sample at $t = 1$ conditioned on $y$ */

---

## 3   Condition-Aware Reparameterization

As detailed in Section 2, standard conditional flow matching initiates the probability path from a condition-agnostic prior, typically a standard Gaussian. Consequently, the velocity network $v_\theta(x_t, t, y)$ must simultaneously learn two intertwined tasks—transporting mass and encoding semantics, imposing a dual burden. To alleviate this, we reparameterize the source and/or target distributions via explicit functions of the condition $y$. In this section, we first reformulate conditional flow-matching using general reparameterizations (Section 3.1). We then show that allowing arbitrary reparameterizations leads to trivial *zero-cost* minima, collapsing distributions to a single mode (Section 3.2). To both shorten the transport path and eliminate these trivial solutions, we propose **Condition-Aware Reparameterization for Flow Matching (CAR-Flow)** (Section 3.3).

### 3.1   General Reparameterization

Formally, instead of drawing an initial value $x_0$ directly from the fixed source distribution $p_x^{\text{init}}$, we first apply a condition-dependent *source distribution map* $f\colon \mathbb{R}^n \times \mathcal{Y} \to \mathbb{R}^m, (x, y) \mapsto z$ such that

$$z_0 = f(x_0, y), \quad x_0 \sim p_x^{\text{init}}. \tag{5}$$

We hence obtain a sample from the modified source distribution $p_z^{\text{init}}$ via the push-forward $z_0 \sim p_z^{\text{init}}(\cdot \mid y) = f(\cdot, y)_\# p_x^{\text{init}}$.

Similarly, we define the *target distribution map* $g\colon \mathbb{R}^n \times \mathcal{Y} \to \mathbb{R}^m, (x, y) \mapsto z$ such that

$$z_1 = g(x_1, y), \quad x_1 \sim p_x^{\text{data}}(\cdot \mid y). \tag{6}$$

We characterize a sample from the *target* distribution in latent space $p_z^{\text{data}}$ via the push-forward $z_1 \sim p_z^{\text{data}}(\cdot \mid y) = g(\cdot, y)_\# p_x^{\text{data}}(\cdot \mid y)$. Critically, to make sampling tractable, $g$ must be approximately invertible—i.e., we assume there exists $g^{-1}$ such that $x_1 \approx g^{-1}(z_1, y)$.

It is worth noting that our reparameterization framework subsumes both the classic VAE-based latent diffusion model [Rombach et al., 2022] and more recent efforts to inject semantic awareness directly into the latent space, e.g., work by Leng et al. [2025]. To recover the standard latent diffusion model, we leave the source untouched, i.e., $f(x_0, y) = x_0$, and set $g(x_1, y) \triangleq \mathcal{E}(x_1), g^{-1}(z_1, y) \triangleq \mathcal{D}(z_1)$, where $\mathcal{E}, \mathcal{D}$ are the encoder and decoder of a VAE trained via the ELBO (reconstruction loss plus KL divergence) without explicit semantic supervision. In contrast, more recent works [Yu et al., 2025, Leng et al., 2025, Yao et al., 2025] augment VAE training with a semantic alignment loss—often using features from a pretrained DINO-V2 network—so that the encoder and decoder effectively depend on semantic embeddings of $y$. Both paradigms arise naturally as special cases of our general push-forward reparameterization framework. Moreover, our framework readily accommodates further reparameterization of any pretrained VAE. Concretely, one can introduce an invertible map $g'\colon \mathbb{R}^m \times \mathcal{Y} \to \mathbb{R}^m$ and set

$$g(x_1, y) \triangleq g'\big(\mathcal{E}(x_1), y\big), \quad g^{-1}(z_1, y) \triangleq \mathcal{D}\big((g')^{-1}(z_1, y)\big), \tag{7}$$

thus embedding semantic conditioning directly into both the encoder and decoder.

**Flow-matching loss and sampling under reparameterization.** When $p_z^{\text{init}}(\cdot \mid y)$ is Gaussian, the probability path $z_0 \to z_1$ remains Gaussian under the interpolation $z_t = \beta_t z_0 + \alpha_t z_1$, $u_t =$

**Algorithm 2** Sampling via conditional flow matching (reparameterized)

---

**Require:** trained $v_\theta$; diffusion schedule $\{\sigma_t\}$; steps $N$
1: Sample $y \sim p_Y$
2: Sample $x_0 \sim \mathcal{N}(0, I_d)$ and set $z \leftarrow f(x_0, y)$
3: $\Delta t \leftarrow 1/N$
4: **for** $k = 0$ **to** $N - 1$ **do**
5:     $t \leftarrow k \, \Delta t$
6:     $u \leftarrow v_\theta(z, t, y)$
7:     Integrate SDE in Eq.(8) over $[t, t + \Delta t]$ with $(u, \sigma_t)$
8: **end for**
9: $x \leftarrow g^{-1}(z, y)$                               /* *invertible g needed here* */
10: **return** $x$                                     /* *sample at $t = 1$ conditioned on $y$* */

---

$\dot{\beta}_t z_0 + \dot{\alpha}_t z_1$. The evolution of $Z_t$ then follows the SDE

$$\mathrm{d}Z_t \;=\; \left[ u_t(Z_t, y) + \frac{\sigma_t^2}{2} \nabla \log p_t(Z_t \mid y) \right] \mathrm{d}t \;+\; \sigma_t \, \mathrm{d}W_t, \tag{8}$$

where $p_t$ is the Gaussian path in $z$-space. Note that the conditional score functions generally differ: $\nabla \log p_t(Z_t \mid y) \neq \nabla \log p_t(X_t \mid y)$, unless $f(x_0, y) = x_0$ (see Appendix A for details). The resulting flow-matching loss becomes

$$\mathcal{L}(\theta) = \mathbb{E}_{\substack{y \sim p_Y, \, t \sim p_T \\ x_0 \sim p_x^{\mathrm{init}}, \, x_1 \sim p_x^{\mathrm{data}}}} \left\| v_\theta\big(\beta_t z_0 + \alpha_t z_1, t, y\big) - \big(\dot{\beta}_t z_0 + \dot{\alpha}_t z_1\big) \right\|^2. \tag{9}$$

Once trained, sampling proceeds by first drawing an initial starting point $x_0 \sim p_x^{\mathrm{init}}$ and computing the condition-aware latent $z_0 = f(x_0, y)$. We then integrate the reparameterized SDE in Eq. (8) from $t = 0$ to $t = 1$ to obtain $z_1$, and finally map back to data space via $x_1 = g^{-1}(z_1, y)$. Algorithm 2 provides a pseudo-code that summarizes the process.

### 3.2 Mode Collapse under Unrestricted Reparameterization

Under the general reparameterization framework of Eq. (9), the maps $f$ and $g$ enjoy full flexibility. Unfortunately, this expressivity also admits several trivial *zero-cost* minima: analytically, the loss can be driven to zero by degenerate shift solutions, resulting in a collapse of the generative distribution to a single/improper mode.

**Claim 1.** *Let $f, g \colon \mathbb{R}^n \times \mathcal{Y} \to \mathbb{R}^m$ be arbitrary maps. If any of the following holds (for some functions $c(y) \in \mathbb{R}^m$ or scalars $k(y) \in \mathbb{R}$):*

  *(i)* Constant source: $f(x_0, y) = c(y)$ *for all $x_0$,*
  *(ii)* Constant target: $g(x_1, y) = c(y)$ *for all $x_1$,*
 *(iii)* Unbounded source scale: $\|f(x_0, y)\| \to \infty$,
 *(iv)* Unbounded target scale: $\|g(x_1, y)\| \to \infty$,
  *(v)* Proportional collapse: $f(x_0, y) = k(y) \, g(x_1, y)$,

*then the flow-matching loss in Eq.* (9) *admits a trivial minimum in which the optimal velocity field takes the form*

$$v_\theta(z_t, t, y) = \gamma(t, y) \, z_t + \eta(t, y),$$

*causing all probability mass to collapse to a single/improper mode.*

The proof of Claim 1, together with closed-form expressions for $\gamma(t, y)$ and $\eta(t, y)$, is deferred to the Appendix B.

To illustrate the collapse concretely, we consider the linear schedule $\beta_t = 1 - t$, $\alpha_t = t$ and an affine reparameterization

$$f(x_0, y) = \sigma_0(y) \, x_0 + \mu_0(y), \qquad g(x_1, y) = \sigma_1(y) \, x_1 + \mu_1(y), \tag{10}$$

analogous to the standard Gaussian trick. Table 1 summarizes each collapse mode, listing the maps, the closed-form collapsed velocity $v_*(z, t, y) = \gamma(t, y) \, z + \eta(t, y)$, and the resulting push-forward distributions $p_z^{\mathrm{init}}$ and $p_z^{\mathrm{data}}$.

Table 1: Zero-cost collapse modes under the linear schedule $\alpha_t = t$, $\beta_t = 1 - t$, using affine maps $f(x_0) = \sigma_0 x_0 + \mu_0$ and $g(x_1) = \sigma_1 x_1 + \mu_1$. For brevity, we omit explicit $y$-dependence.

| Case | $f$ | $g$ | $\gamma(t)$ | $\eta(t)$ | $v_*(z_t, t)$ | $p_z^{\text{init}}$ | $p_z^{\text{data}}$ |
|------|-----|-----|-------------|-----------|---------------|---------------------|---------------------|
| (i) | $\mu_0$ | arbitrary | $\frac{1}{t}$ | $-\frac{1}{t}\mu_0$ | $\frac{z_t - \mu_0}{t}$ | $\delta_{\mu_0}$ | – |
| (ii) | arbitrary | $\mu_1$ | $-\frac{1}{1-t}$ | $\frac{1}{1-t}\mu_1$ | $\frac{\mu_1 - z_t}{1-t}$ | – | $\delta_{\mu_1}$ |
| (iii) | $\infty$ | arbitrary | $-\frac{1}{1-t}$ | $0$ | $-\frac{z_t}{1-t}$ | $\text{Uniform}(\mathbb{R}^d)$ | – |
| (iv) | arbitrary | $\infty$ | $\frac{1}{t}$ | $0$ | $\frac{z_t}{t}$ | – | $\text{Uniform}(\mathbb{R}^d)$ |
| (v) | $\mu_0$ | $k\mu_0$ | $0$ | $(k-1)\mu_0$ | $(k-1)\mu_0$ | $\delta_{\mu_0}$ | $\delta_{\mu_1}$ |

We note that in the *constant-map* scenarios (cases (i) and (ii)), the affine transforms $f$ or $g$ effectively collapse the variance of $p_z^{\text{init}}$ or $p_z^{\text{data}}$ to zero. In particular, case (ii) mirrors the finding in REPA-E (Table 1) [Leng et al., 2025], where end-to-end VAE–diffusion tuning favored a simpler latent space with reduced variance. In contrast, the *unbounded-scale* modes (iii) and (iv) push the distributions toward an improper uniform distribution, eliminating any meaningful localization. *Proportional collapse* (v) yields a degenerate flow with $p_z^{\text{init}}$, $p_z^{\text{data}}$, and the velocity field being constant.

Empirically, we do not observe cases (iii)–(v). In particular, cases (iii) and (iv) correspond to *unbounded collapse* solutions: when the scale of either the source or target distribution tends to infinity, the counterpart distribution collapses relative to it, yielding zero cost. These solutions require unbounded weights and are therefore unstable—any perturbation pushes the optimization back toward cases (i) or (ii). Case (v), in contrast, assumes exact proportionality between maps $f$ and $g$, which cannot occur under independently sampled inputs except in trivial constant-map settings that reduce to cases (i) or (ii). In practice, the optimizer thus follows the easiest "shortcut"—the *constant-map* collapse—since setting $f$ or $g$ to a fixed constant immediately zeroes the loss. In Section 4, we empirically verify not only that such constant-map collapses occur in real models, but also how these mode-collapse behaviors manifest in practice.

## 3.3 Shift-Only Condition-Aware Reparameterization

To eliminate the trivial zero-cost solutions while retaining the benefits of condition-aware reparameterization, we restrict both reparameterizations to *additive shifts* only, while noting that volume-preserving maps are possible. For simplicity we use

$$f(x_0, y) = x_0 + \mu_0(y), \qquad g(x_1, y) = x_1 + \mu_1(y). \tag{11}$$

Here, $\mu_0, \mu_1 \colon \mathcal{Y} \to \mathbb{R}^m$ are lightweight, learnable, condition-dependent shifts. By preserving scale, we block every collapse mode given in Claim 1, yet we still move $z_0$ and $z_1$ closer in latent space. In particular, when used with a pretrained VAE, Eq. (7) becomes

$$g(x_1, y) \triangleq \mathcal{E}(x_1) + \mu_0(y), \quad g^{-1}(z_1, y) \triangleq \mathcal{D}(z_1 - \mu_1(y)), \tag{12}$$

CAR-Flow admits three natural variants:

- **Source-only:** $\mu_1 \equiv 0$, so only the source is shifted.
- **Target-only:** $\mu_0 \equiv 0$, so only the target is shifted.
- **Joint:** both $\mu_0$ and $\mu_1$ active.

Each variant alters the probability path $z_t = \beta_t (x_0 + \mu_0(y)) + \alpha_t (x_1 + \mu_1(y))$ in a distinct way. Note that shifting the source cannot be replicated by an opposite shift on the target. In fact, no nonzero constant shift on the source can ever be matched by a constant shift on the target—except in the trivial case:

**Claim 2.** *Shifting the source by $\mu_0$ is equivalent to shifting the target by $\mu_1$, if and only if $\mu_0 = \mu_1 = 0$.*

*Proof.* With a source-only shift $(\mu_0, \mu_1) = (\mu_0, 0)$, the interpolant is $z_t^{(s)} = \beta_t (x_0 + \mu_0) + \alpha_t x_1$. With a target shift $(\mu_0, \mu_1) = (0, \mu_1)$, it is $z_t^{(t)} = \beta_t x_0 + \alpha_t (x_1 + \mu_1)$. Equating these gives for $\forall t$, $\beta_t \mu_0 = \alpha_t \mu_1$. Since $\mu_0$ and $\mu_1$ are $t$-independent functions of $y$, it forces $\mu_0 = \mu_1 = 0$. □

For **Source-only**, the interpolant simplifies to $z_t = x_t + \beta_t \mu_0(y)$. When $\mu_0(y) \approx \mu_0(y')$ for two conditions $y, y'$, their paths share a common starting region, simplifying the early-time flow. For

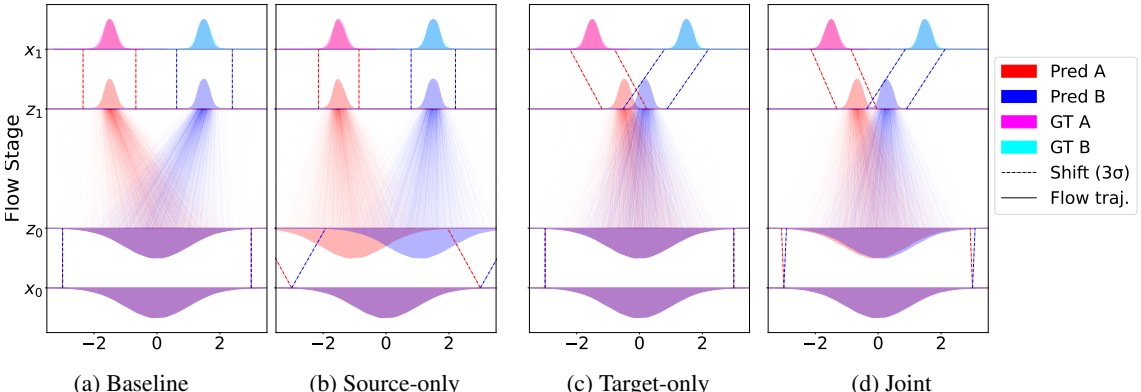

Figure 2: **Learned flow trajectories on 1D synthetic data.** Each panel shows trajectories from source $x_0$ (bottom) to target $x_1$ (top) for (a) baseline and CAR-Flow variants–(b) source-only, (c) target-only, and (d) joint. Intermediate stages $z_0$ and $z_1$ reflect reparameterized coordinates. Colored densities represent predicted and ground-truth class distributions (red/blue: prediction; magenta/cyan: ground truth). Thin lines illustrate individual sample trajectories between $z_0$ and $z_1$. Dashed vertical lines mark $\pm 3\sigma$ for each shift. The source-only CAR-Flow relocates the source distribution per class, while the target-only variant unifies the trajectory endpoints. The joint variant combines both and achieves the best alignment and flow quality.

**Target-only**, the trajectory becomes $z_t = x_t + \alpha_t \, \mu_1(y)$. When $\mu_1(y) \approx \mu_1(y')$, the network only needs to learn a shared "landing zone", easing the late-time flow. For **Joint**, we have $z_t = x_t + \mu_t(y)$ where $\mu_t(y) = \beta_t \, \mu_0(y) + \alpha_t \mu_1(y)$, so the time-varying shift aligns both endpoints, minimizing the overall transport distance and reducing the burden on $v_\theta$ throughout the entire trajectory.

Empirically, we find that the joint CAR-Flow variant—allowing both $\mu_0$ and $\mu_1$ to adapt—yields the largest improvements in convergence speed and sample fidelity (see Section 4).

## 4 Experiments

In this section, we evaluate the efficacy of the Condition-Aware Reparameterization for Flow Matching (CAR-Flow) under a linear noise schedule $\beta_t = 1 - t$, $\alpha_t = t$ and compare to classic rectified flow. All experiments are done using the axlearn framework.[2] Detailed implementation settings can be found in Appendix C.

### 4.1 Synthetic Data

For our synthetic-data experiments, we consider a one-dimensional task where the source distribution is $\mathcal{N}(0, 1)$ and the target distribution is a two-class Gaussian mixture, with class A data distribution $\mathcal{N}(-1.5, 0.2^2)$ and class B data distribution $\mathcal{N}(+1.5, 0.2^2)$.

We encode $x_t$, $y$, and $t$ using sinusoidal embeddings before feeding them to the network. In the baseline rectified-flow model, these embeddings are concatenated and passed through a three-layer MLP (1,993 parameters total) to predict the velocity field $v_\theta(x_t, y, t)$. Training uses the loss in Eq. (4), and sampling follows Algorithm 1. For CAR-Flow, we augment this backbone with two lightweight linear layers that map the class embedding to the shifts $\mu_0(y)$ and/or $\mu_1(y)$, each adding only 9 parameters. In the source-only variant we predict $\mu_0$ (with $\mu_1 = 0$); in the target-only variant we predict $\mu_1$ (with $\mu_0 = 0$); and in the joint variant we predict both. These shifts are applied via Eq. (11), training proceeds with the loss in Eq. (9), and sampling uses Algorithm 2. All models—baseline and CAR variants—set $\sigma_t = 0$ (reducing the SDE to an ODE) and employ a 50-step Euler integrator for sampling. To ensure robustness, each configuration is run three times, and we report the average performance, noting that variance across runs is negligible.

Figure 2 shows how CAR-Flow alters the learned flow. In Figure 2a, the baseline must both transport mass and encode class information, yielding the longest paths. The source-shift variant in Figure 2b

---

[2]`https://github.com/apple/axlearn`

Table 2: Average trajectory length $\|z_0 \to z_1\|$ with $2\sigma$ error bounds.

| | Baseline | Source-only CAR-Flow | Target-only CAR-Flow | Joint CAR-Flow |
|---|---|---|---|---|
| Length | $1.5355 \pm 0.0024$ | $0.7432 \pm 0.0019$ | $0.7129 \pm 0.0010$ | $0.7121 \pm 0.0011$ |

(a) Wasserstein distance

(b) Learned $\mu_0$ and $\mu_1$ shifts

Figure 3: **Comparison of convergence and learned shifts.** (a) shows the Wasserstein distance between predicted and ground-truth distributions in symlog-scale. Joint CAR-Flow achieves both the fastest convergence (b) plots the evolution of the learned shifts $\mu_0$ (top) and $\mu_1$ (bottom) for two classes.

cleanly relocates each class's start, the target-shift variant in Figure 2c leaves the source untouched and merges endpoints, and the joint variant in 2d aligns both start and end with minimal source shift—producing the shortest trajectories (Table 2).

Figure 3a summarizes convergence measured by Wasserstein distance over training: joint CAR-Flow converges fastest and reaches the lowest error, followed by source and target-only, all outperforming the baseline. Figure 3b traces the learned $\mu_0$ and $\mu_1$ for each class and variant: joint CAR-Flow yields the most moderate, balanced shifts, explaining its superior convergence and flow quality.

**Mode Collapse.** To empirically validate the mode-collapse analysis described in Section 3.2, we reuse the setup but now allow both shift and scale parameters to be learned simultaneously (Eq.(10)). We train two separate models: one reparameterizing the source distribution with learned parameters $(\mu_0, \sigma_0)$, and another morphing the target distribution with $(\mu_1, \sigma_1)$. Results are presented in Figure 4. Specifically, Figure 4a shows the rapid evolution of the learned standard deviations $\sigma$, clearly indicating the network quickly discovers a "shortcut" solution by shrinking $\sigma$ to zero. Figure 4b plots the expected norm gap $\mathbb{E}\|v_\theta - v_*\|^2$, demonstrating convergence to zero, which indicates the network's velocity prediction aligns closely with the analytic *zero-cost* solutions derived in Table 1. Moreover, unrestricted parameterization of the source distribution triggers mode-collapse case (i) as described in Claim 1, where the flow degenerates to predicting the class-wise mean of the target (see Figure 4c). Conversely, unrestricted parameterization of the target collapses the distribution nearly to a constant, and consequently, the predicted $x_1$ degenerates into an improper uniform distribution due to $\sigma_1 \to 0$, as illustrated in Figure 4d.

### 4.2 ImageNet

To benchmark on a high-dimensional, large-scale dataset, we conduct experiments on ImageNet $256 \times 256$ data using v6e-256 TPUs. Our baseline is SiT-XL/2 [Ma et al., 2024], re-implemented in JAX [Bradbury et al., 2018]; we strictly follow the original training recipe from the open-source SiT repository to replicate the results reported in the paper. For our CAR variants, we apply the shift-only reparameterization to the *sd-vae-ft-ema* VAE backbone used by SiT (see Eq. (11)- (12)). We introduce two lightweight convolutional networks ($\approx 2.3M$ parameters each) to predict $\mu_0$ and $\mu_1$ from the class embeddings, projecting them into the latent space. All models are sampled using the Heun SDE solver with 250 NFEs.

Table 3 presents the quantitative results. Augmenting SiT-XL/2 with CAR-Flow consistently outperforms the baseline across all variants. In particular, the joint-shift variant achieves the best result, reducing FID from 2.07 to 1.68 while adding fewer than $0.6\%$ parameters. These results underscore the importance of explicitly conditioning the source and target distributions: simple shift reparam-

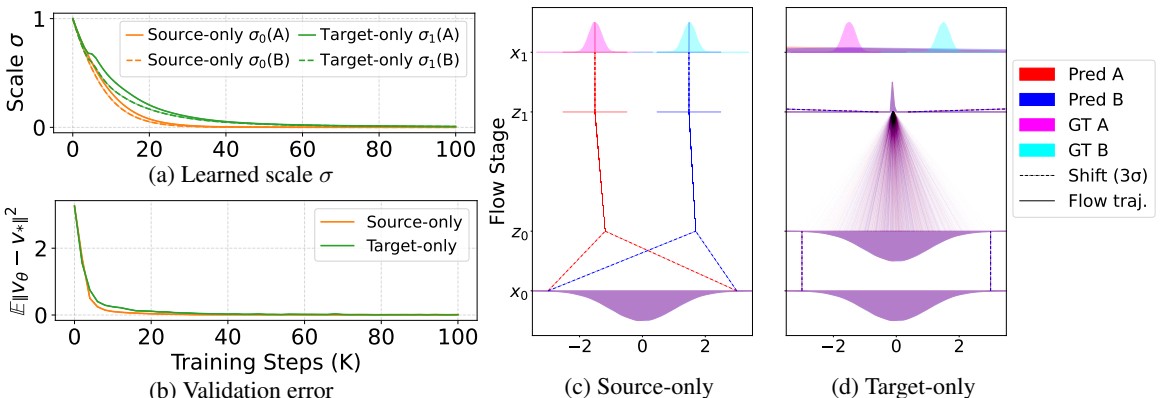

Figure 4: **Mode collapse diagnostics with scale reparameterization.** (a)–(b) Evolution of learned $\sigma$ and validation error. (c)–(d) Learned flows when allowing shift+scale on source vs. target.

Table 3: Benchmarking class-conditional image generation on ImageNet $256 \times 256$. All CAR-Flow variants surpass the SiT-XL/2 baseline.

| Method | Params (M) | Training Steps | FID ↓ | IS ↑ | sFID ↓ | Precision ↑ | Recall ↑ |
|---|---|---|---|---|---|---|---|
| SiT-XL/2 | 675 | 400K | 17.28 | 78.88 | 5.71 | 0.67 | 0.61 |
| SiT-XL/2$_{\text{CAR-Flow Source-only}}$ | 677 | 400K | 15.15 | 83.42 | 5.56 | 0.68 | 0.61 |
| SiT-XL/2$_{\text{CAR-Flow Target-only}}$ | 677 | 400K | 14.44 | 85.66 | 5.59 | 0.68 | 0.61 |
| SiT-XL/2$_{\text{CAR-Flow Joint}}$ | 679 | 400K | **13.91** | **87.96** | **5.38** | **0.68** | **0.62** |
| SiT-XL/2$_{\text{cfg=1.5}}$ | 675 | 7M | 2.07 | 280.2 | 4.46 | 0.81 | 0.58 |
| SiT-XL/2$_{\text{CAR-Flow Joint+cfg=1.5}}$ | 679 | 7M | **1.68** | **304.0** | **4.34** | **0.82** | **0.62** |

eterization not only improves sample fidelity but does so with minimal computational overhead, facilitating easy integration into existing large-scale generative frameworks.

Table 3 presents the quantitative results. Augmenting SiT-XL/2 with CAR-Flow consistently outperforms the baseline across all variants. In particular, the joint-shift variant achieves the best result, reducing FID from 2.07 to 1.68 while adding fewer than $0.6\%$ parameters. Beyond final performance, CAR-Flow also accelerates optimization: our convergence analysis on ImageNet-256 shows that all CAR-Flow variants consistently reduce FID faster than the baseline across training steps (see Fig. 5). These results underscore the importance of explicitly conditioning the source and target distributions: simple shift reparameterization not only improves sample fidelity but does so with minimal computational overhead, facilitating easy integration into existing large-scale generative frameworks.

## 5 Related Work

Generative modeling has advanced significantly in the last decade from variational auto-encoders (VAEs) [Kingma and Welling, 2014], Generative Adversarial Nets [Goodfellow et al., 2014], and normalizing flows [Rezende and Mohamed, 2015]. More recently, score matching [Song and Ermon, 2019, Song et al., 2020], diffusion models [Ho et al., 2020], and flow matching [Liu et al., 2023, Lipman et al., 2023, Albergo and Vanden-Eijnden, 2023, Albergo et al., 2023] was introduced. The latter three frameworks are related: sampling at test-time can be viewed as numerically solving a transport (ordinary) differential equation by integrating along a learned velocity field from the source distribution at time zero to the target distribution at time one.

For learning the velocity field, various approaches to interpolate between samples from the source distribution and the target distribution have been discussed [Lipman et al., 2023, Liu et al., 2023,

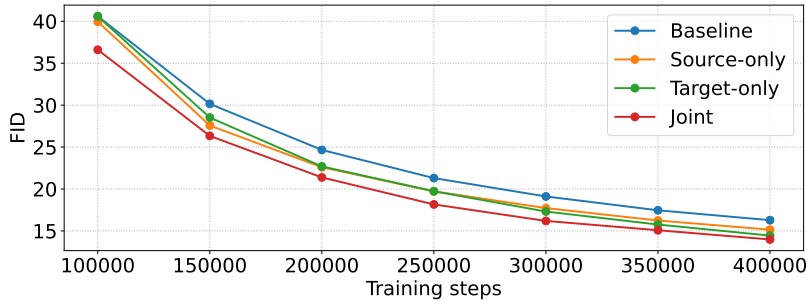

Figure 5: Convergence on ImageNet $256 \times 256$: FID vs. training steps. CAR-Flow variants consistently converge faster than the baseline.

Tong et al., 2024]. Among those, rectified flow matching was shown to lead to compelling results on large-scale data [Ma et al., 2024, Esser et al., 2024].

For use on large-scale data, flow matching is typically formulated in latent space by compressing data via the encoder of a pre-trained and frozen VAE [Rombach et al., 2022]. These mappings differ from the discussed CAR-Flow, as they are typically independent of the conditioning variable. As mentioned before, CAR-Flow can be applied on top of pre-trained and frozen projections on the latent space.

More recently, Yu et al. [2025], Yao et al. [2025] proposed to align representations within deep nets that model the velocity to visual representations from vision foundation models. Specifically, Yu et al. [2025] align early layer features of DiT [Peebles and Xie, 2023] and SiT [Ma et al., 2024] models with representations extracted from DINOv2 [Oquab et al., 2024] and CLIP [Radford et al., 2021]. In contrast, Yao et al. [2025] aligns the latent space of a VAE with representations from pre-trained vision foundation models, which are then frozen for diffusion model training. These approaches differ from our approach, which learns to transform the source and target distributions rather than encouraging feature alignment.

Most related to our work is the recently introduced REPA-E [Leng et al., 2025]. In REPA-E, Leng et al. [2025] study end-to-end training of diffusion models and VAE encoders/decoders, which map data to/from a latent space. Differently, in this paper, we formalize failure modes reported by Leng et al. [2025] and identify them as trivial solutions that arise when jointly training a flow matching model and a target distribution mapping. We further introduce a source distribution mapping. Finally, we impose simple restrictions that preclude those trivial solutions.

# 6 Conclusion

We propose and study condition-aware reparameterization for flow matching (CAR-Flow), which aligns the source and target distributions in flow-matching models. We find CAR-Flow to alleviate the burden of classic flow-matching, where a model simultaneously transports probability mass to the correct region of the data manifold, while also encoding the semantic meaning of the condition.

**Limitations and broader impact.** This work characterizes the failure modes when jointly training a flow matching model, a source distribution mapping, and a target distribution mapping, and identifies them as trivial solutions of the objective. We further study the simplest effective approach to avoid these trivial solutions. While we find this simple approach to lead to compelling results, we think more general mappings will likely improve results even further. We leave the identification of more general suitable mappings to future work.

Improving the expressivity of generative models has a significant broader impact. On the positive side, modeling complex distributions more easily saves resources and enables novel applications. On the negative side, efficient generative modeling can be abused to spread misinformation more easily.

## Acknowledgements

We begin by thanking Byeongjoo Ahn, Wenze Hu, Zhe Gan, and Jason Ren for their thoughtful discussions and rapid feedback, which significantly shaped this study. We also acknowledge the Apple Foundation Model team for providing the critical infrastructure that powered our experiments Finally, we are profoundly appreciative of Ruoming Pang and Yang Zhao for their strategic vision, steadfast guidance, and unwavering encouragement throughout this project.

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

# Appendix — CAR-Flow: Condition-Aware Reparameterization Aligns Source and Target for Better Flow Matching

The appendix is organized as follows:

- In Section A we derive the effects of our reparameterization on the score function. This is important for correct sampling, particularly when using an SDE solver.
- In Section B we prove Claim 1.
- In Section C we provide additional implementation details for both synthetic and ImageNet experiments.
- In Section D we discuss some findings.
- In Section F we illustrate additional qualitative results.

## A  Score Function Under Reparameterization

In this section we show that

1. The conditional score $s_t = \nabla \log p(z_t \mid y)$ generally *changes*, once the source-space map $f$ is applied, unless $f(x, y) = x$;
2. For the practically important *shift-only* map $f(x, y) = x + \mu_0(y)$ under a Gaussian path, both the density and the score admit closed forms; and
3. The score couples to the drift (velocity field) $u_t$ that appears in the SDE used for sampling.

### A.1  Change of the Conditional Score

Recall that the source transform is $f \colon \mathbb{R}^n \times \mathcal{Y} \to \mathbb{R}^m, (x, y) \mapsto z$. Let $J_f(x, y) \in \mathbb{R}^{m \times n}$ denote its Jacobian. By the change-of-variables formula,

$$p_X(x \mid y) = p_Z\big(f(x, y) \mid y\big) \big|\det J_f(x, y)\big|. \tag{13}$$

Taking $\nabla_x \log$ of Eq. (13) gives

$$\boxed{\nabla_x \log p_X(x \mid y) = J_f(x, y)^\top \underbrace{\nabla_z \log p_Z\big(f(x, y) \mid y\big)}_{\text{score in } z\text{-space}} + \nabla_x \log\big|\det J_f(x, y)\big|.} \tag{14}$$

The first term transports the $z$-space score back to the $x$-space tangent via the Jacobian transpose; the second term corrects for the local volume change introduced by $f$.

**Volume-preserving maps.** If $f$ is volume preserving, $\det J_f \equiv \pm 1$ and hence $\nabla_x \log |\det J_f| = 0$. Eq. (14) then reduces to $\nabla_x \log p_X = J_f^\top \nabla_z \log p_Z$. In the special case of $f(x, y) = x$, one has $J_f = I$ and therefore $\nabla_x \log p_X = \nabla_x \log p_Z$, recovering the classical setting *without* reparameterization.

### A.2  Shift-only Transform

Assume the initial distribution $p_x^{\text{init}} = \mathcal{N}(0, I_d)$ is standard Gaussian and $f$ is a shift-only map, i.e., $f(x, y) = x + \mu_0(y)$. Along a Gaussian path parameterized by $\alpha_t, \beta_t$ (with boundary conditions $\alpha_0 = 0, \beta_0 = 1$ and $\alpha_1 = 1, \beta_1 = 0$), the conditional density at time $t$ is

$$p_t(\cdot \mid z_1, y) = \mathcal{N}\big(\alpha_t z_1 + \beta_t \mu_0, \beta_t^2 I_d \mid y\big), \tag{15}$$

with both endpoints being

$$p_0(\cdot \mid z_1, y) = \mathcal{N}\big(\mu_0, I_d \mid y\big), \qquad p_1(\cdot \mid z_1, y) = \delta_{z_1 \mid y}. \tag{16}$$

Consequently, by using the form of the Gaussian probability density, the (conditional) score reads

$$s_t(z_t \mid z_1, y) = \nabla_{z_t} \log p_t(z_t \mid z_1, y) = \frac{\alpha_t z_1 + \beta_t \mu_0(y) - z_t}{\beta_t^2}. \tag{17}$$

### A.3 Link Between Score and Velocity Field

Fix a conditioning label $y$ and a pair $(z_0, z_1)$ drawn from the Gaussian endpoint distributions introduced in Section. 3.1. The flow is defined as

$$\psi_t(z_0 \mid z_1, y) \; = \; \beta_t\, z_0 + \alpha_t\, z_1, \qquad 0 \le t \le 1, \tag{18}$$

so that $\psi_0 = z_0$ and $\psi_1 = z_1$. Because $\psi_t$ satisfies the ODE $\mathrm{d}\psi_t / \mathrm{d}t = u_t\big(\psi_t \mid z_1, y\big)$, differentiating Eq. (18) in $t$ gives the *conditional* velocity field

$$u_t(\alpha_t z_1 + \beta_t z_0 \mid z_1, y) \; = \; \dot\beta_t\, z_0 + \dot\alpha_t\, z_1. \tag{19}$$

Since $z_t = \psi_t(z_0 \mid z_1, y)$, $z_0 = (z_t - \alpha_t z_1)/\beta_t$, substituting these into Eq. (19) yields

$$u_t(z_t \mid z_1, y) \; = \; \dot\beta_t \left( \frac{z_t - \alpha_t z_1}{\beta_t} \right) + \dot\alpha_t\, z_1. \tag{20}$$

Eq. (17) links $z_1$ and the *conditional* score $s_t(z_t \mid z_1, y)$. Solving Eq. (17) for $z_1$ and inserting the result into Eq. (20) yields

$$u_t(z_t \mid z_1, y) = \frac{\dot\alpha_t}{\alpha_t} z_t + \left( \beta_t^2 \frac{\dot\alpha_t}{\alpha_t} - \dot\beta_t \beta_t \right) \left( \underbrace{\frac{\alpha_t z_1 + \beta_t \mu_0(y) - z_t}{\beta_t^2}}_{s_t(z_t \mid z_1, y)} - \underbrace{\frac{\mu_0(y)}{\beta_t}}_{\text{bias correction}} \right). \tag{21}$$

The first term under the brace is precisely the *conditional score* from Eq. (17), while the second term compensates for the mean shift $\mu_0(y)$ introduced by the shift-only transformation.

Eq. (21) can be rearranged to express the score through the velocity:

$$s_t(z_t \mid z_1, y) = \frac{\alpha_t\, u_t(z_t \mid z_1, y) - \dot\alpha_t\, z_t}{\beta_t^2 \dot\alpha_t - \alpha_t \dot\beta_t \beta_t} + \frac{\mu_0(y)}{\beta_t}. \tag{22}$$

To translate the conditional identity given in Eq. (22) to the *marginal* setting used at inference time, we integrate over the target endpoint $z_1$. For this we introduce the posterior

$$q_t(z_1 \mid z_t, y) \; := \; \frac{p(z_t \mid z_1, y)\, p_z^{\text{data}}(z_1 \mid y)}{p_t(z_t \mid y)}, \qquad \text{with} \qquad \int q_t(z_1 \mid z_t, y)\, \mathrm{d}z_1 = 1.$$

The marginal velocity and score are simply expectations under this density, i.e.,

$$u_t(z_t \mid y) = \mathbb{E}_{q_t}\big[u_t(z_t \mid z_1, y)\big], \qquad s_t(z_t \mid y) = \mathbb{E}_{q_t}\big[s_t(z_t \mid z_1, y)\big].$$

Applying Eq. (22) inside the expectation and using linearity yields

$$\boxed{s_t(z_t \mid y) \; = \; \frac{\alpha_t\, u_t(z_t \mid y) - \dot\alpha_t\, z_t}{\beta_t^2 \dot\alpha_t - \alpha_t \dot\beta_t \beta_t} \; + \; \frac{\mu_0(y)}{\beta_t}}, \tag{23}$$

the exact coupling used in the latent-space SDE (Eq. 8) to express the score with the drift during sampling.

## B  Proof of Claim 1

In this section we prove Claim 1 in two steps: (i) We show that *any* parameter choice driving the objective in Eq. (9) to its global minimum forces the velocity field to be affine in the interpolant, $v_\theta(z_t, t, y) = \gamma(t, y)\, z_t + \eta(t, y)$. (ii) For each collapse pattern (i)–(v) in the claim, we verify that the loss indeed attains this trivial minimum and we state the resulting $\gamma(t, y)$ and $\eta(t, y)$ in closed form.

**Step 1. Affine Form at Zero Loss**

Let $z_t = \beta_t z_0 + \alpha_t z_1$. If the loss in Eq. (9) vanishes almost surely, the integrand must be identically zero:

$$v_\theta(z_t, t, y) = \dot\beta_t\, z_0 + \dot\alpha_t\, z_1 \quad \forall(z_0, z_1, t, y). \tag{24}$$

Fix $(t, y)$ and apply the chain rule w.r.t. $z_0$ and $z_1$:

$$\frac{\partial v_\theta}{\partial z_t} \beta_t = \dot{\beta}_t, \qquad \frac{\partial v_\theta}{\partial z_t} \alpha_t = \dot{\alpha}_t.$$

The right–hand sides are constant in $(z_0, z_1)$, hence $\partial v_\theta / \partial z_t \equiv \gamma(t, y)$ does not depend on $z_t$. Integrating once in $z_t$ gives the *affine* form

$$v_\theta(z_t, t, y) = \gamma(t, y)\, z_t + \eta(t, y), \tag{25}$$

with $\eta(t, y)$ an integration constant. Thus any zero-loss solution necessarily has the affine form given in Eq. (25) and is already independent of $\theta$. This completes Step 1.

**Step 2. Each Collapse Pattern Attains the Zero–loss Affine Field**

Write $z_0 := f(x_0, y)$ and $z_1 := g(x_1, y)$. Insert the ansatz given in Eq. (25) into the pointwise identity given in Eq. (24) and substitute $z_t = \beta_t z_0 + \alpha_t z_1$:

$$\gamma(t, y)\big[\beta_t z_0 + \alpha_t z_1\big] + \eta(t, y) = \dot{\beta}_t\, z_0 + \dot{\alpha}_t\, z_1. \tag{26}$$

Eq. (26) is linear in $(z_0, z_1)$; each collapse scenario reduces it to one or two scalar conditions, from which $\gamma$ and $\eta$ are obtained explicitly.

(i) **Constant source.** $z_0 \equiv c(y)$ is fixed while $z_1$ varies freely. Matching the $z_1$-coefficient in Eq. (26) forces $\gamma = \dot{\alpha}_t / \alpha_t$. The remaining scalar equation fixes $\eta = c(y)\,(\dot{\beta}_t - \gamma\beta_t)$.

(ii) **Constant target.** Symmetric to case (i): $\gamma = \dot{\beta}_t / \beta_t$ and $\eta = c(y)\,(\dot{\alpha}_t - \gamma\alpha_t)$.

(iii) **Unbounded source scale.** As $\|z_0\| \to \infty$ with $z_1$ bounded, the $z_0$-terms in Eq. (26) dominate; finiteness of the left-hand side requires $\gamma\beta_t = \dot{\beta}_t \Rightarrow \gamma = \dot{\beta}_t / \beta_t$. With this choice the entire identity holds for *all* $(z_0, z_1)$ when we set $\eta = 0$.

(iv) **Unbounded target scale.** Analogous to (iii) with roles exchanged: $\gamma = \dot{\alpha}_t / \alpha_t$ and $\eta = 0$.

(v) **Proportional collapse.** Suppose $z_0 = k(y)z_1$. Substituting into Eq. (26) yields a single free variable $z_1$: $\gamma\big[\beta_t k(y) + \alpha_t\big] z_1 = \big[\dot{\beta}_t k(y) + \dot{\alpha}_t\big] z_1$. Hence $\gamma(t, y) = (\dot{\beta}_t k(y) + \dot{\alpha}_t)/(\beta_t k(y) + \alpha_t)$ and $\eta(t, y) = 0$.

In all five situations $\gamma$ and $\eta$ depend only on $(t, y)$ and the collapse maps $(f, g)$. Consequently the optimizer can reach a *trivial* minimum in which $v_\theta$ no longer guides a meaningful flow and the generated distribution collapses to a single/improper mode. $\qquad\square$

## C   Implementation Details

### C.1   Synthetic Data

The velocity network consumes three sinusoidal position embeddings that encode the latent state $x_t$, the class label $y$, and time $t$. Each embedding has dimensionality 8, and concatenating them yields a 24-dimensional feature vector. This vector is processed by a three-layer MLP whose hidden layers are all $24 \to 24$ linear projections followed by GELU activations. A final linear layer maps $24 \to 1$, producing the scalar velocity. The entire model—including all embedding parameters—contains 1 993 trainable parameters.

To implement the additive shifts $\mu_0(y)$ and $\mu_1(y)$ we introduce two lightweight condition networks. Each consists of a single linear layer that maps the 8-dimensional class embedding to a scalar shift, plus a bias term, for 9 parameters per network. Both linear layers are initialized with all weights and biases set to *zero*, ensuring the additive shifts are identically zero at the start of training.

We train with a batch size of 1 024 using AdamW with $\beta_1 = 0.9$ and $\beta_2 = 0.95$. Learning rates are fine-tuned per parameter group: $1 \times 10^{-3}$ for the source shift network, $1 \times 10^{-4}$ for the target shift network, and $1 \times 10^{-5}$ for all remaining parameters. Unless noted otherwise, models are trained for 50k steps; mode-collapse experiments are extended to 100k steps to ensure convergence.

All the synthetic data experiments were executed on the CPU cores of an Apple M1 Pro laptop.

## C.2   ImageNet

We re-implemented the open-source SiT[3] code-base in JAX and reproduced the SiT/XL-2 configuration on ImageNet at $256 \times 256$ resolution as our baseline. The exact architecture, datapipeline, optimizer (AdamW) and learning-rate schedule are identical to the original code. Training is performed on a single v6e-256 TPU slice.

For both source and target CAR-Flows, we append a lightweight label-conditioning network that maps the 1152-dimensional class embedding to a latent tensor of shape $32 \times 32 \times 4$:

- Dense: $1152 \rightarrow 128 \times 4 \times 4$.
- Upsampling: three repeats of $\mathrm{ConvTranspose2d(kernel\_size{=}2, stride{=}2)} \rightarrow \mathrm{GroupNorm} \rightarrow$ ReLU; shapes $4 \times 4 \times 128 \rightarrow 8 \times 8 \times 64 \rightarrow 16 \times 16 \times 32 \rightarrow 32 \times 32 \times 16$.
- Head: $3 \times 3$ convolution (padding 1) to $32 \times 32 \times 4$, initialized with all weights and bias set to *zero* to ensure the no shifts at the start of training.

Each network contains 2.4M parameters ($\sim 0.3\%$ of the SiT/XL-2 backbone) incurring negligible overhead.

We inherit the original AdamW optimizer for the SiT backbone with learning rate of $1 \times 10^{-4}$ and global batch size of 256. Both label-conditioning networks are trained with a higher learning rate $1 \times 10^{-1}$; all other hyper-parameters are unchanged.

# D   Discussion of Findings

## D.1   Relative Learning Rates

During our experiments, we discovered that the relative learning rate between the lightweight condition networks (which learn the additive shifts $\mu_0(y)$ and $\mu_1(y)$) and the velocity-network backbone has a first-order effect on both the magnitude of the learned shifts and the convergence speed—a "race condition" between them.

To study this, we reuse the synthetic-data example: we fix the backbone's learning rate at $1 \times 10^{-5}$ and train source-only and target-only CAR-Flow models while sweeping the shift-network learning rate across three orders of magnitude ($1 \times 10^{-5}$, $1 \times 10^{-4}$, and $1 \times 10^{-3}$). Figures 6a and 6c plot the $\mu_0$ and $\mu_1$ trajectories for the source-only and target-only variants, respectively. At the smallest rate, the shifts remain near zero and the backbone carries the conditioning, yielding slower alignment; at the intermediate rate, the shifts grow steadily—though they can still be pulled by the backbone (e.g., $\mu_0(y = B)$ starts positive but gradually becomes negative, reducing $|\mu_0(y = A) - \mu_0(y = B)|$); and at the largest rate, the shifts rapidly attain larger magnitudes and drive the quickest convergence. Figures 6b and 6d report the Wasserstein distances, confirming that the highest shift-network rate achieves the fastest distribution alignment.

These observations suggest that empowering the shift networks with a higher learning rate can substantially accelerate alignment. In practice, one should choose a shift-network rate that is sufficiently high to speed convergence while preserving robust inter-class separation.

To verify that these trends extend to large-scale data, we performed the same learning-rate sweep on ImageNet using the SiT-XL/2 backbone. We fixed the backbone's learning rate at its default $1 \times 10^{-4}$ and trained the CAR-Flow condition networks under three configurations—source-only, target-only, and joint—while varying their learning rate from $1 \times 10^{-3}$ to $1 \times 10^{-1}$. Figure 7 shows the FID at 400K steps for each variant, with a SiT-XL/2 baseline of 17.2 indicated by the dashed line.

At the lower condition-network rate ($\sim 10^{-3}$), source-only and target-only variants remain on-par or worse than the baseline, whereas the joint variant already outperforms it. As the rate increases, all three variants deliver substantial improvements—target-only achieves the best FID of 12.77 at $2 \times 10^{-2}$, and the joint variant exhibits the most stable performance across the sweep. Even at the highest rate ($10^{-1}$), all configurations remain well below the baseline.

These large-scale results mirror our synthetic-data findings: increasing the shift-network learning rate accelerates alignment and improves sample quality, although the exact optimum varies by variant and requires some tuning. For simplicity and robust performance, we therefore adopt a rate of $1 \times 10^{-1}$

---

[3]`https://github.com/willisma/SiT`

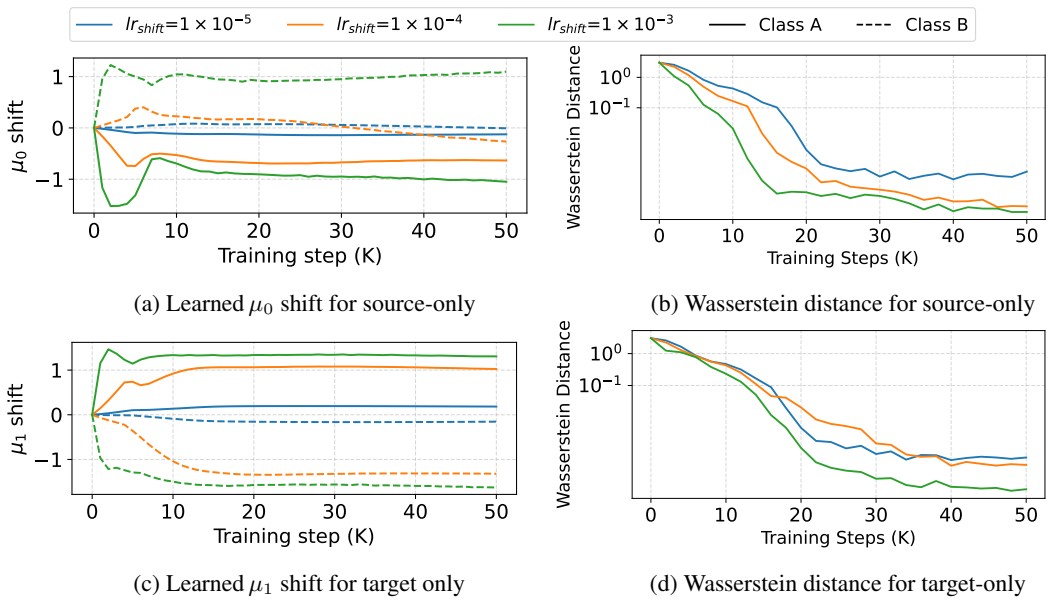

(a) Learned $\mu_0$ shift for source-only

(b) Wasserstein distance for source-only

(c) Learned $\mu_1$ shift for target only

(d) Wasserstein distance for target-only

Figure 6: Effect of varying the shift-network learning rate relative to a fixed backbone rate of $1 \times 10^{-5}$. Panels (a) and (c) show the trajectories of $\mu_0$ and $\mu_1$ for three learning rates; panels (b) and (d) plot the corresponding 1-D Wasserstein distances. At the lowest shift-network rate, the learned shifts remain negligible (slow alignment); at the intermediate rate, they grow steadily without instability; and at the highest rate, they overshoot then damp, yielding the fastest overall convergence despite early oscillations.

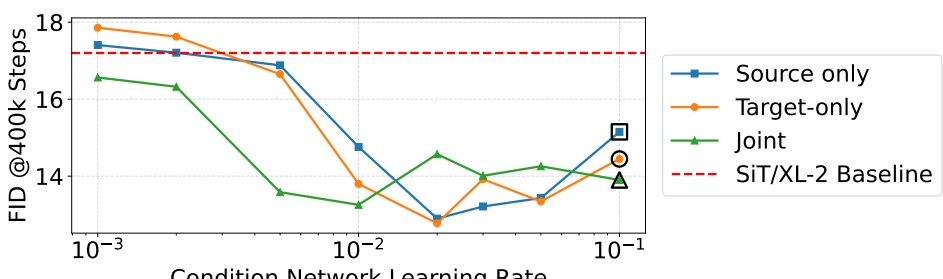

Figure 7: ImageNet FID at 400K steps vs. condition-network learning rate for SiT-XL/2 CAR-Flow variants: source-only (blue), target-only (orange), and joint (green). The dashed horizontal line marks the baseline FID of 17.2. All variants improve over the baseline as the shift-network rate increases.

in our main experiments. Figure 8 presents example outputs from each CAR-Flow variant at this rate, demonstrating that the joint variant attains superior visual fidelity compared to the source-only and target-only models.

## D.2 Condition-aware source versus unconditional shift

Complementary to the observations above, we examine whether making the condition-aware is beneficial compared to using a single unconditional shift. For intuition, consider the source-only CAR-Flow variant: it reparameterizes the source with a shift that depends on $y$. When the target distribution varies with the conditioning variable, aligning the source per condition should reduce transport effort relative to an unconditional source [Albergo et al., 2023].

We test this in the 1-D synthetic setup of Sec. 4.1 by comparing (i) a learnable unconditional source with a global shift and (ii) a condition-aware source with a $y$-dependent shift. The Wasserstein distances are $0.058$ for (i) versus $0.041$ for (ii), indicating that per-condition alignment reduces transport. This supports CAR-Flow's design choice and clarifies its distinction from an unconditional "learnable source".

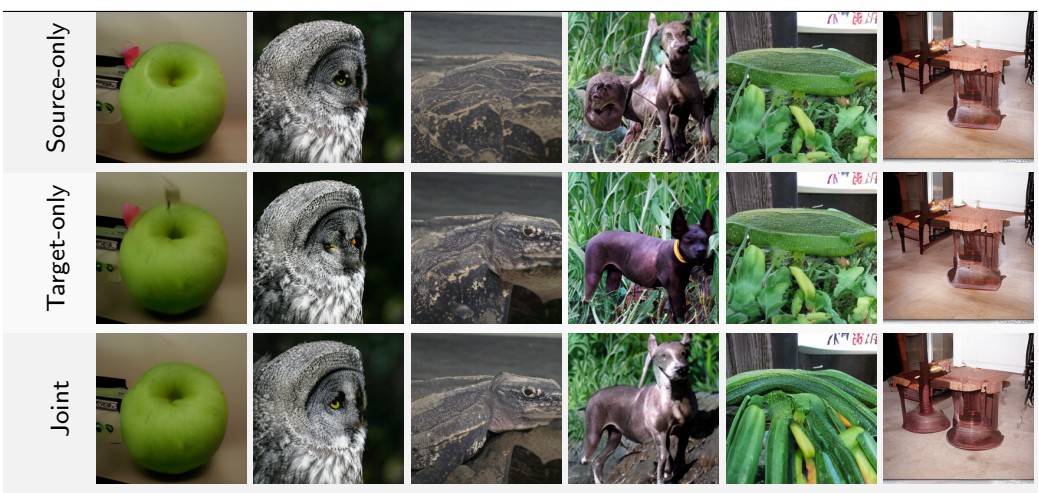

Figure 8: Qualitative ablation of CAR-Flow variants on SiT-XL/2 at 400K steps. Each row corresponds to one variant—(top) source-only, (middle) target-only, and (bottom) joint CAR-Flow—and each column shows a generated sample for a different class from the same noise using $cfg = 1$. The joint model produces the most realistic and semantically accurate images across all scenarios.

Table 4: Class-conditional image generation on CIFAR-10 (FID ↓).

|  | Baseline | Source-only CAR-Flow | Target-only CAR-Flow | Joint CAR-Flow |
|---|---|---|---|---|
| **FID** ↓ | 13.8 | 7.5 | 11.1 | 10.6 |

# E    Additional Results on CIFAR-10

To assess generalization beyond ImageNet, we trained a SiT-XL/2 baseline and CAR-Flow variants for 400k steps on CIFAR-10 using pixel-space diffusion (VAE omitted due to CIFAR-10's low resolution $32 \times 32$). All CAR-Flow variants outperformed the baseline, demonstrating that the benefits of CAR-Flow generalize across datasets.

# F  Qualitative Results

Qualitative results are provided in Figure 9.

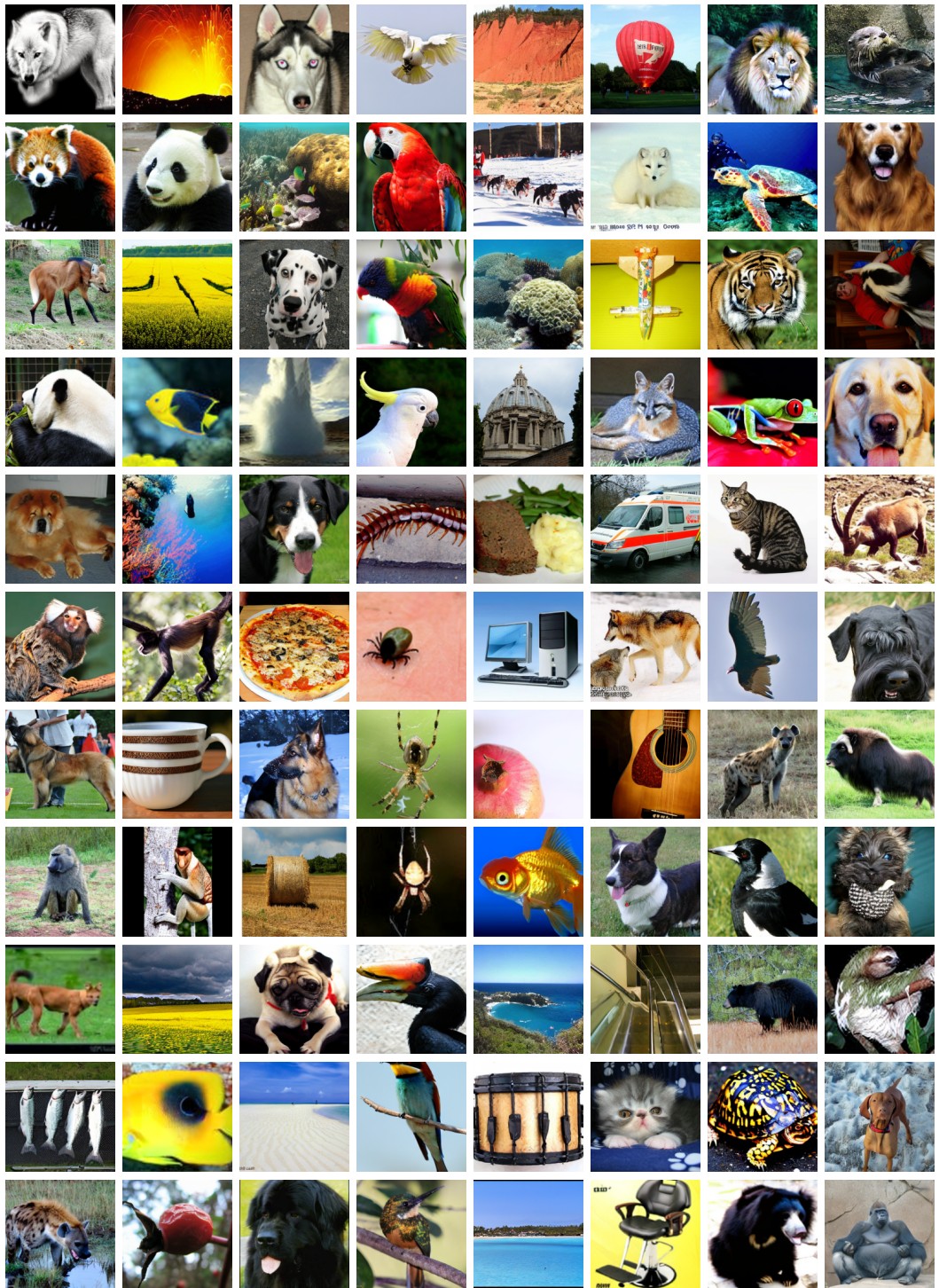

Figure 9: Randomly selected samples generated from our SiT-XL/2$_{\text{CAR-Flow Joint}}$ model trained on ImageNet $256 \times 256$ data using $cfg = 4$.

