# OpenReview forum: "CAR-Flow: Condition-Aware Reparameterization Aligns Source and Target for Better Flow Matching"
_NeurIPS.cc/2025/Conference — NeurIPS 2025 spotlight_

### Official Review · Reviewer_zCmc · 2025-06-20

**Clarity:** 4
**Significance:** 3
**Originality:** 3
**Rating:** 5
**Confidence:** 3

**Summary:**

This work introduces CAR, a technique to improve conditional flow matching models by learning a transformation of the latent and/or data distribution that depends on the condition. The flow matching model is then applied to map between these two transformed distributions. More specifically, the authors first identify five classes of degenerate zero-cost solutions of flow matching that can be achieved when using arbitrary transformations. They then notice that it is impossible to create these solutions with only a shift of both distributions, and thus propose to include a learned shift before and after the mapping induced by flow matching. They then verify in a toy example that the proposed shift transformation learns a non-degenerate solution, whereas including scaling also leads to one of the zero-cost solutions in practice. Finally, they demonstrate that even this simple addition to the FM framework yields significant improvements on conditional ImageNet 256x256.

**Questions:**

- Do you have a notion of how complete the list of failure modes in Claim 1 is? Are there other problems beyond mode collapse that could occur?
- Have you tried to apply CAR for continuous conditions? Can you observe similar benefits?
- Have you tried more general transformations (e.g. volume-preserving transformations would seem to avoid all the trivial solutions as well)?

**Ethical Concerns:**

["NO or VERY MINOR ethics concerns only"]

**Final Justification:**

This paper presents a simple, universally applicable method to improve the conditional flow matching framework. Being one of the most relevant frameworks in conditional generative modelling, this method can be expected to have a significant impact on the field of generative modelling as a whole, which is supported by solid (but not exhaustive) theoretical and experimental evaluation. I do not rate this work higher (i.e. groundbreaking) as it only considers a first analysis of the simplest settings (one type of data/condition in the experiments, only the simplest transformation that avoids the identified trivial solutions).

**Limitations:**

yes

**Quality:**

3

**Strengths And Weaknesses:**

**Strengths**
- The presented method is simple and seems applicable to most conditional flow matching scenarios
- The results achieved on ImageNet 256x256 are very convincing
- The identified trivial minima are not only relevant for this work but might help to illuminate problems that occur in adjacent work which performs joint training of flow matching and pre-/post-hoc transformations.
- The method is very lightweight and seems to have almost no implementational and computational overhead
- The writing is very clear, proofs are formulated understandably and the toy example is quite instructive.

**Weaknesses**
- The method's quantitative and qualitative performance was only compared against baseline flow matching on a single dataset. While the results are convincing there is still room to explore how it performs for different dataset and condition types
- The improvements in convergence speed were only shown for the 1-d toy example. It would be interesting to see if improved convergence speed is also observed on high-dimensional data.
- The authors only consider the simplest transformation that avoids the identified trivial solutions. More complex transformations might result in much better performance

Minor things:
- The qualitative results in the appendix lack a baseline comparison
- The shift prediction network learning rate needs to be fine-tuned to see the beneficial effects of CAR (but according to Fig. 6 it seems very lenient)

In conclusion, while the extent of performance gain that can be expected from CAR (and possibly more advanced versions) might not be quantified entirely from the limited experiments in this paper, due to the mentioned strengths it can be expected to overall contribute positively to CFM training.

---

> ### Author Rebuttal · Authors · 2025-07-31
>
> Thank you for your time and thoughtful feedback. We’re excited to see that the reviewer found our method “applicable to most conditional flow matching scenarios,” our ImageNet results “very convincing,” our identified trivial solutions broadly relevant, and our approach both lightweight and clearly presented. We address the reviewer’s questions and suggestions below.
>
> ----
>
> - $\textbf{Reviewer comment (W1)}$: Performance on additional datasets.
>
> Thank you for the suggestion. To assess generalization beyond ImageNet, we conducted an additional experiment on CIFAR-10. We trained a SiT/XL-2 baseline and CAR variants for 400k steps, using pixel-space diffusion (omitting the VAE due to CIFAR-10’s resolution). Results are shown below:
>
> | Model | FID |
> |-------------------|------|
> | Baseline | 13.8 |
> | Source-only CAR | 7.5 |
> | Target-only CAR | 11.1 |
> | Joint CAR | 10.6 |
>
>
> All CAR variants outperformed the baseline, demonstrating that the benefits of CAR generalize on different datasets.
>
> ----
>
> - $\textbf{Reviewer comment (W2)}$: Convergence speed on high-dimensional data.
>
> Thank you for pointing this out. We conducted a new convergence analysis on ImageNet-256 using the SiT-XL/2 backbone. The table below reports FID over training steps, showing that CAR variants consistently converge faster than the baseline:
>
>
> | **Steps** | **Baseline** | **Source-only** | **Target-only** | **Joint** |
> |-----------|--------------|-----------------|-----------------|-----------|
> | 50k | 75.76 | 74.44 | 74.11 | 71.18 |
> | 100k | 40.60 | 39.96 | 40.59 | 36.60 |
> | 150k | 30.15 | 27.57 | 28.52 | 26.33 |
> | 200k | 24.66 | 22.59 | 22.69 | 21.38 |
> | 250k | 21.30 | 19.72 | 19.73 | 18.16 |
> | 300k | 19.11 | 17.72 | 17.31 | 16.19 |
> | 350k | 17.46 | 16.24 | 15.76 | 15.08 |
> | 400k | 16.28 | 15.15 | 14.44 | 13.98 |
>
>
> We will add this table (and the corresponding learning curves) to the revision to document CAR’s convergence advantage on high-dimensional data.
>
>
> ----
>
> - $\textbf{Reviewer comment (W3/Q3)}$: Exploring more complex transformations.
>
> Thank you for raising this excellent point. We focused on shift-only reparameterizations as they are the simplest volume-preserving transforms that already yield measurable benefits with minimal overhead. We agree that richer—but still volume-preserving—mappings (e.g., learned orthonormal transforms) could potentially deliver further improvements while avoiding the trivial minima identified in Claim 1. Systematically exploring these more expressive options is a promising next step.
>
> ----
>
> - $\textbf{Reviewer comment (W3)$: Missing baseline samples in qualitative results.
>
> We will add side-by-side qualitative comparisons with the baseline in the appendix of the revised manuscript. Although differences are subtle at later training stages (e.g., 7M steps with classifier-free guidance), they are more noticeable earlier in training. For instance, CAR-generated Chihuahua samples at 400k steps (without classifier-free guidance) exhibit stronger structural coherence than those from the baseline. Due to the text-only rebuttal format, we cannot include images here, but we will make them available in the final version.
>
> ----
>
> - $\textbf{Reviewer comment (W4)}$: Lenience to shift-prediction network learning rate.
>
> Indeed, in practice we found CAR settings to be quite lenient: its gains hold across a wide span of shift-network learning rates, so only coarse tuning is needed.
>
> ----
>
> - $\textbf{Reviewer comment (Q1)}$: Completeness of failure modes in Claim 1.
>
> Thank you for the question. Cases (iii)–(v) in Claim 1 are theoretical degenerate “collapse” solutions that can be derived from Eq. (26) in the Appendix. As noted in L164–165, they require extreme conditions that do not arise in practice. While additional failure collapse modes may exist in theory, we believe the practically relevant ones are already captured by cases (i) and (ii), which we observe empirically (see Fig. 4). In practice, beyond these mode-collapse scenarios, we have not encountered other failure modes in our experiments.
>
> ----
>
> - $\textbf{Reviewer comment (Q2)}$: Applicability to continuous conditions.
>
> We have not yet run experiments with continuous‐valued conditions. Conceptually, CAR should extend straightforwardly—for instance, by modeling $P(\mu | y)$ where  $y$ is continuous. We expect similar benefits in such settings, though verifying this would require additional experimentation beyond the scope of this submission.

---

> > ### Comment · Reviewer_zCmc · 2025-08-04
> >
> > Thank you for answering my questions and providing additional results.
> > These support my positive impression of the paper and I will keep my current rating.

---

> > > ### Author Response · Authors · 2025-08-04
> > >
> > > Thank you for your thoughtful feedback and for taking the time to review our work. We’re glad the additional results helped clarify our contributions, and we appreciate your positive assessment and continued support.

---

### Official Review · Reviewer_n8K3 · 2025-07-01

**Clarity:** 3
**Significance:** 3
**Originality:** 3
**Rating:** 4
**Confidence:** 4

**Summary:**

The paper proposes CAR, condition-aware reparametrization, a technique to train source and target transformations simultaniously with the velocity field in the flow matching frameworks. This approach allocates some capacity for processing the conditioning separately from the main flow matching objective, which fosters the alignment between the source and the target distributions, and thereby eases the optimization of the velocity field. The authors provide theoretical analysis of the degenerate solutions in the general unresticted settings and derive the shift-only CAR to avoid those. Experiments on synthetic and real-world data support the claims made in the paper and demonstrate superior performance compared to the prior work.

**Questions:**

1. While the method is motivated for the conditional case, I think the unconditional generation can also benefit from the proposed source/target alignment. Could the authors provide any insights on why the proposed approach should/sould not work for the unconditional setting?

**Ethical Concerns:**

["NO or VERY MINOR ethics concerns only"]

**Final Justification:**

The paper addresses an interesting problem of jointly training source and target transformations along with the velocity field in flow matching models. The proposed approach and results appear sound. The rebuttal addressed most of my concerns. However, I believe the clarity of the theoretical section could still be improved (in line with the discussion with the authors), hence my rating.

**Limitations:**

yes

**Paper Formatting Concerns:**

Figures 2 and 4 and Table 3 appear to be slightly off the right margin.

**Quality:**

3

**Strengths And Weaknesses:**

### Strengths
1. The paper is well-written and the motivation is clear. The method is simple and theoretical foundations are provided. The results on both synthetic and real data support the claims made in the paper.

### Weaknesses

1. The main weakness of the paper is the lack of clarity in delivering the theoretical results. It is not clear why the degenerate 0-cost solutions are presented as "$\theta$-independent". These are analytical solutions that exist in the general setting, but if the $\theta$-depended model converged to these during the optimization process they still depend on parameters.
2. The proportional collapse case is unclear. In the loss (equation 9), $x_0$ and $x_1$ are sampled independently from each other. However, case *(v)* somehow suggests that $z_0$ and $z_1$ are coupled. Could the authors clarify this?
3. The unbounded case also requires further clarifications. In general, the 0-cost solution of the optimization problem in equation 9 is feasible if the are no intersections between the trajectories that interpolate $z_0$ and $z_1$. While in cases *(i)* and *(ii)* this is clear, in the case where both the target and the source distributions have non-trivial supports, there will be intersections, unless the source and the target are non-trivially coupled. I would like to see more discussion of this issue in the cases *(iii)-(v)*.

---

> ### Author Rebuttal · Authors · 2025-07-31
>
> Thanks a lot for your time and valuable feedback. We appreciate that the reviewer finds the paper to be "well-written and the motivation is clear". We are glad to see that the reviewer appreciates the provided theoretical foundations and acknowledges that the results on both synthetic and real data support our claims. Below we address each of the reviewer's comments and questions in turn.
>
> ----
>
> - $\textbf{Reviewer comment (W1)}$: Clarify what is meant by "$\theta$-independent" degenerate solutions.
>
> Thank you for highlighting this confusion. We agree that our phrasing was unclear. Our intention was to indicate that the degenerate solutions $v^*$ yield zero cost analytically—by construction—when paired with corresponding source/target shifts. However, the learned velocity field $v_{\theta}$ must still approximate this ill-conditioned optimum via parameter $\theta$. In that sense, the solution is not truly $\theta$-independent. We will revise the wording accordingly in the final version.
>
> ----
>
> - $\textbf{Reviewer comment (W2/W3)}$: Clarify the unbounded collapse cases and the proportional collapse case.
>
> We appreciate the opportunity to clarify. Cases (iii)–(v) in Claim 1 describe purely theoretical collapse solutions derived from Eq. (26) in the Appendix:
>
> 1) Cases (iii) and (iv) (unbounded collapse): When the scale of either the source or target distribution tends to infinity, the counterpart distribution collapses relative to it. This leads to minimal trajectory overlap between $z_0=f(x_0)$ and $z_1=g(x_1)$, resulting in near-zero cost.
>
> 2) Case (v) (proportional collapse): This case assumes exact proportionality between $z_0$ and $z_1$. As the reviewer correctly notes, this cannot occur when $x_0$ and $x_1$ are sampled independently—except in trivial cases where both $z_0$ and $z_1$ are constants, which reduces to cases (i) or (ii).
>
> As discussed in L164–165, these collapse modes do not arise in practice. Cases (iii) and (iv) require unbounded weights, and are therefore unstable—any  perturbation pushes optimization causes the optimization to revert to cases (i) or (ii). Case (v) requires implausible, perfectly aligned reparameterizations under independently sampled inputs. We included these cases for theoretical completeness and will revise the manuscript to clarify their pathological nature.
>
> ----
>
> - $\textbf{Reviewer comment (Q1)}$: Can the proposed method benefit unconditional generation as well?
>
> Yes--thank you for raising this point. We believe the same source/target reparameterization can benefit unconditional settings. Even without explicit conditioning, allowing learned shifts in the source and/or target distributions gives the model more flexibility to reduce transport cost, thereby simplifying the velocity field it must learn. A related idea—adjusting the prior to ease conditional transport—was briefly mentioned by Albergo & Vanden-Eijnden (2023), but not studied in detail.

---

> > ### Comment · Reviewer_n8K3 · 2025-08-02
> > **Response to the Rebuttal**
> >
> > Thank you for the detailed response. I appreciate the clarifications made in the rebuttal. I have also checked the other reviews. It seems that the unbounded and the proportional degenerate cases have also caused some confusion for the other reviewers. The authors admit that the proportional case can only happen in practice as a special case of (i) or (ii). Thus, it doesn't make sense to me to consider it as a separate case. Regarding the unbounded case, it remains counterintuitive for me and the authors claim near-zero cost in that case in the rebuttal, while the theorem in the paper is about zero-cost solutions. That said, I still appreaciate the method and its performance in the experiments, but I would suggest reformulating the theoretical part. I will hence keep my original rating.

---

> > > ### Author Response · Authors · 2025-08-05
> > >
> > > Thanks a lot for your thoughtful comments:
> > >
> > > - On the proportional case: We would like to clarify that this case can arise in more general frameworks like OT-CFM, where $x_0$​, $x_1$ ​ are sampled jointly from $\pi(x_0, x_1)$, making it clearly different from cases (i) and (ii). We'll clarify this distinction in the revision.
> > >
> > > - Regarding the unbounded case: our theoretical result shows that the zero-cost solution is theoretically achievable when $f/g$ maps to infinity. Practically, if at all possible, deep nets typically only approach such a solution rather than attaining it, hence our phrasing of "near-zero cost" in the rebuttal. We'll clarify.
> > >
> > > Please also note that we already cautioned that cases (iii)-(v) are not practical in L164-165 by stating:
> > >
> > > > Setting maps $f$ or $g$ to a fixed constant immediately zeroes the loss, whereas driving scales to infinity or enforcing precise proportionality would require extreme weights or perfect alignment between maps f and g that standard architectures and training rarely meet.
> > >
> > > We'll make this more explicit in the final version.

---

> > > > ### Comment · Reviewer_n8K3 · 2025-08-06
> > > > **Response to the Authors' comment**
> > > >
> > > > I appreciate the authors' further clarifications:
> > > >
> > > > - Regarding the proportional case, even with OT-CFM it is achievable only in ideal settings (i.e. when $\pi(x_0, x_1)$ is a 1-to-1 assignment). In general, however, OT-CFM only approximates the OT plan by calculating OT per-batch, which constitutes a non-trivial coupling distribution, where the proportional case cannot arise.

---

> > > > > ### Author Response · Authors · 2025-08-06
> > > > >
> > > > > Thank you again for your thoughtful feedback and for engaging deeply with our theoretical discussion. As previously noted, we do acknowledge that cases (iii)–(v) are not practical in typical training regimes (see L164–165), and we will make this point more explicit in the final version.
> > > > >
> > > > > While we agree that these cases may only arise in idealized settings, we believe that discussing such edge cases contributes to a more complete theoretical understanding. Even if impractical, they can help the community better identify potential failure modes and inform model design.

---

> > > > > > ### Comment · Reviewer_n8K3 · 2025-08-07
> > > > > > **Response to the Authors’ comment**
> > > > > >
> > > > > > I appreciate the authors’ willingness to update the paper and improve the clarity of the theoretical section. One possible option could be to move the discussion of impractical cases to the supplementary material, which I think would help the main text flow better. But the authors can decide what works best for them. That said, I am inclined to keep my original positive rating.

---

### Official Review · Reviewer_rueW · 2025-07-03

**Clarity:** 3
**Significance:** 4
**Originality:** 3
**Rating:** 5
**Confidence:** 3

**Summary:**

The authors in this paper propose Condition-Aware Reparameterization (CAR) as a method for imprving conditional generative models especially in diffusion and flow-matching frameworks. Standard conditional generation methods rely on a standard Gaussian prior, forcing the model to handle both semantic conditioning and distribution matching, leading to inefficient, long probability paths during training. Proposed method mitigates this by shifting the source and/or target distributions in a condition-aware manner aligning distributions to reduce the burden on the main model. The authors provide theoretical analysis demonstrating that unrestricted reparameterizations can cause mode collapse. By restricting to additive shifts only, CAR avoids these trivial minima while maintaining benefits in path shortening. Experiments on synthetic data and ImageNet-256 show that CAR consistently improves convergence speed and sample fidelity.

**Questions:**

1. How does CAR perform when conditioning signals are noisy or weak?
2. Have you explored combining CAR with various transformations under constraints?
3.Can you discuss how this will extend to models like FFJORD or neural autoregressive flows?
4.Some ablation study results will be appreciated.
5. Can CAR be extended to multi-modal or multi-task settings?

**Ethical Concerns:**

["NO or VERY MINOR ethics concerns only"]

**Final Justification:**

Based on the authors rebuttal I am upgrading my score.

**Limitations:**

The authors can discuss the limitations in some more detail, covering multi modal modeling, more flow models especially like FFJORD and neural autoregressive flows, cases when conditioning signal is weak, transformation under constraints, etc.

**Paper Formatting Concerns:**

No concerns

**Quality:**

3

**Strengths And Weaknesses:**

Strengths:
1. The paper proposes an interesting solution for reducing the complexity of conditional flow matching by aligning distributions via learnable shifts.
2. The theoretical analysis of the paper is well complimented by empirical analysis.
3. CAR is easy to integrate into existing frameworks, adding minimal overhead, and provides consistent improvements in convergence and sample quality.

Weaknesses:
1. The method is currently limited to shift-only reparameterizations.
2. Detailed ablation study should be performed to quantify contribution of various components.
3. Empirical validation focuses on FID and ImageNet; more diverse datasets and modalities (e.g., text, audio) are not explored.
4. How CAR works for models such as FFJORD or neural autoregressive flows is yet to be covered. (Grathwohl, Will, et al. "Ffjord: Free-form continuous dynamics for scalable reversible generative models." arXiv preprint arXiv:1810.01367 (2018)., Huang, Chin-Wei, et al. "Neural autoregressive flows." International conference on machine learning. PMLR, 2018.)

---

> ### Author Rebuttal · Authors · 2025-07-31
>
> Thanks a lot for your time and valuable feedback. We are glad the reviewer finds our solution to be “interesting”, the “theoretical analysis of the paper” to be “well complemented by empirical analysis”, and our method to be “easy to integrate into existing frameworks” while “adding minimal overhead” and while leading to “consistent improvements in convergence and sample quality”.  Below we address each of the reviewer's comments and questions in turn.
>
>
> ----
>
> - $\textbf{Reviewer comment (W1)}$: Can CAR go beyond shift-only reparameterizations?
>
> We appreciate the suggestion. CAR focuses on shift-only reparameterizations because they are the simplest volume-preserving transforms. Allowing more expressive transformations—especially ones that can change volume—introduces the risk of degenerate solutions (as characterized in Claim 1). By constraining ourselves to shifts, we sidestep such instability while still observing strong empirical gains with negligible overhead. Exploring richer yet stable, volume-preserving reparameterizations is an exciting direction for future work.
>
> ----
> - $\textbf{Reviewer comment (W2/Q4)}$: Request for ablation studies.
>
> In the current manuscript, we have already conducted ablations to isolate CAR’s components:
>
> 1) CAR variants (Source-only, target-only, and joint CAR): Fig. 2 & Table 2 (synthetic data); Table 3 (ImageNet).
>
> 2) Learning-rate ablations: Appendix Fig. 5 (synthetic data) and Fig. 6 (ImageNet).
>
> If there are additional components the reviewer would like us to examine, please let us know. We are happy to incorporate any further ablations in the revision.
>
>
>
> ----
> - $\textbf{Reviewer comment (W3/Q5)}$: Limited dataset diversity—only ImageNet/FID.
>
> Thank you for this suggestion. While ImageNet remains a widely used benchmark for high-dimensional generation, we additionally ran a new ablation on CIFAR-10 to assess generality.
>
> We trained a SiT/XL-2 baseline and CAR variants for 400k steps, using pixel-space diffusion (VAE omitted due to CIFAR-10’s resolution). Results:
>
>
> | Model            | FID  |
> |-------------------|------|
> | Baseline          | 13.8 |
> | Source-only CAR   | 7.5  |
> | Target-only CAR   | 11.1 |
> | Joint CAR         | 10.6 |
>
>
>
> All CAR variants outperform the baseline, indicating generalization beyond ImageNet. We believe CAR can also extend to other modalities (e.g., text, audio), but compute constraints prevent us from including such experiments in this submission.
>
> ----
> - $\textbf{Reviewer comment (W4/Q3)}$: Can CAR be applied to FFJORD or neural autoregressive flows?
>
> We see no fundamental barrier to applying CAR in frameworks like FFJORD or neural autoregressive flows. However, each new setting may introduce its own degenerate collapse modes (analogous to the ones characterized in Claim 1), and a detailed theoretical and empirical study is required to identify and address those cases.
>
> ----
> - $\textbf{Reviewer comment (Q1)}$: How does CAR perform under weak or noisy conditioning?
>
> In conditional flow-matching models, when the conditioning signal $y$ is weak or noisy, both the baseline and CAR-augmented models are affected—there is simply less reliable information to guide the learned velocity field $v_{\theta(x_t, t, y)}$. This limitation is not unique to our approach but is inherent to all conditional generative methods. Improving robustness to weak or noisy conditioning is an interesting but orthogonal direction to our contribution.

---

> > ### Comment · Reviewer_rueW · 2025-08-05
> >
> > Thank you for the response, these answer my questions, and I have upgraded my score.

---

### Official Review · Reviewer_naoP · 2025-07-03

**Clarity:** 3
**Significance:** 3
**Originality:** 2
**Rating:** 4
**Confidence:** 2

**Summary:**

* The paper introduces Condition-Aware Reparameterization (CAR) which is a new way of doing conditional generative modeling. CAR explicitly incorporates conditioning information into generative models via learned, shift-only reparameterizations of either the source or target distributions. This simplifies the training of flow-based generative models and enhances their sampling efficiency.

* The authors provide theoretical insights, demonstrating that allowing certain families of reparameterizations can lead to degenerate, trivial solutions. By restricting reparameterizations to shift-only transformations, CAR avoids these degenerate cases.

* In the synthetic dataset example, trajectory lengths being shorter does not necessarily imply that generation quality improves. Indeed, the CAR method can be seen as decomposing the trajectory into different stages x_0 -> z_0 -> z_1 -> x_1, so naturally the length of the z_0 -> z_1 path would be shorter. However, by measuring the Wasserstein divergence of the generated distribution and the ground truth, the authors verify that the quality of the generated samples is better when using CAR.

* Empirical validation shows CAR reduces trajectory lengths and improves sampling quality. Specifically, it achieves gains on ImageNet generation benchmarks (e.g., improving FID from 2.07 to 1.68 on ImageNet-256) with minimal additional parameters.

**Questions:**

* Do you have any examples (theoretical or empirical) where learning the distribution for y performs better than just learning it for x_0? (also see first weakness above)
* You mention that your method decouples the “transporting mass” and “encoding semantics.” However, I don’t see any reason for it being problematic to learn these two behaviors at the same time. Is there any intuition you could provide for what you have in mind here?
* Do you have any intuition on why the target-only CAR leads to paths with smaller lengths and lower FIDs than the source-only CAR?
* The meaning of cases (iii) and (iv) is not very clear from what is written in Claim 1.

**Ethical Concerns:**

["NO or VERY MINOR ethics concerns only"]

**Final Justification:**

The authors did a good job clearing up a few questions and confusions I had about the paper. Thanks to the authors' answer I have updated the significance score and the overall score.

**Limitations:**

Yes

**Quality:**

2

**Strengths And Weaknesses:**

Strengths
* The authors do a good gob laying out the foundations and formalizing their framework. The presentation overall is mostly clear.
* The paper provides both a simple demonstration of their proposed algorithm (CAR) on synthetic dataset. This is useful to understand that CAR reduces the trajectory length in latent space, potentially leading to less error due to discretization. On the other hand, the authors also show that CAR improves FID performance on ImageNet, which is evidence of the practical use of this method.

Weaknesses
* Although the theoretical claims (Claim 1 and Claim 2) are interesting, they are mostly characterizing degenerate behavior. This leaves the reader without a sense of what are optimal \mu_0 and \mu_1 in non-degenerate cases. Understanding theoretically the case where the target is a two-class Gaussian mixture would be ineresting. If that is too hard, a simplification would be to understand the optimal \mu_0 for the case where the target is a symmetric sum of diracs (which I believe doesn’t match the triviality conditions from Claim 1)
* It is known in the literature that changing the source distribution can lead to lower transport costs (e.g. Albergo and Vanden-Eijnden 2023). Instead of learning a distribution for the y, we could just learn a distribution for the x_0 instead of  using the N(0, Id). Note that there are ways of doing this using only a few parameters like still parameterizing x_0 by a Gaussian, but learning the mean and covariance (where we could assume the covariance is diagonal if we want to save parameters.) Although this wouldn’t decouple “transporting mass” and “encoding semantics”, it is not clear that this is needed, and having a learnable distribution for x_0 may already explain the performance increase.

---

> ### Author Rebuttal · Authors · 2025-07-31
>
> Thanks a lot for your time and valuable feedback. We appreciate the reviewer highlighting that the paper does “a good job laying out the foundations and formalizing the framework”. We are glad to see that the reviewer appreciates that we show a combination of both simple demonstrations and practical impact. Below we address each of the reviewer’s comments and questions in turn.
>
> ----
>
> - $\textbf{Reviewer comment (W1)}$: Claims 1 & 2 mostly rule out degenerate cases; they don’t tell us what the optimal $\mu_0$, $\mu_1$ are in non-degenerate cases.
>
> We appreciate this insightful observation. We would like to point out that there is generally no unique optimal $\mu_0$, $\mu_1$. Instead they live in a family of equally valid solutions because they are jointly learned with the velocity field $v_{\theta}$.
>
> Consider the simplest 1-D example, where the source and target are Dirac masses at $x_0$ and $x_1$. In the case of CAR (with source shift learned), the optimal velocity is $v^*=x1-(x0+\mu_0)$. Hence, for any choice of $\mu_0$, there exists a corresponding zero-cost velocity field. Once training finds any member of this family, the gradient on $\mu_0$ vanishes. In practice, its final value is governed by many factors (learning-rate, model architecture, flow complexity, etc.).
> We will clarify this point and include the example above in the revision.
>
> ----
>
> - $\textbf{Reviewer comment (W2/Q1)}$: Why not simply learn an unconditional source distribution $p(x_0)$ (Albergo & Vanden-Eijnden, 2023)? Do you have any evidence that learning the distribution for $y$ is actually better?
>
> To make sure, note that we don’t learn a distribution for the condition $y$; rather, we learn a re-parameterized source that depends on the condition $y$. Intuitively, when the target distribution varies with the conditioning variable, aligning the source per condition should reduce transport effort beyond what a single unconditional source can offer.
>
> To validate this, we ran a controlled experiment using the Sec. 4.1 1-D synthetic experiment setup that compares
>
> (i) a learnable $\textbf{unconditional}$ source $p(x_0)$ with shift,
>
> (ii) a $\textbf{condition-dependent}$ source $p(x_0\mid y)$ with shift dependent on y.
>
> We got Wasserstein distance of (i) 0.058 vs. (ii) 0.041. This supports our claim: aligning the source per condition leads to reduced transport effort. We will add this result to the final version and clarify the distinction between an unconditional “learnable source” and CAR’s condition-aware variant.
>
> ----
>
> - $\textbf{Reviewer comment (Q2)}$: Intuition for learning “transporting mass” and “encoding semantics”?
>
> Thank you for this question. We do not mean to imply that a single velocity field $v_{\theta}(x_t, t, y)$ cannot learn both mass transport and semantic injection simultaneously. Rather, our intuition and goal is to simplify what the flow is asked to model:
> 1) In standard flow matching, a single, condition-agnostic source must be carried to potentially very different target manifolds for each $y$. As a result, the network must learn both long-range transport and the precise semantic shifts encoded by $y$, which can lead to intricate trajectories and slower convergence.
> 2) In CAR, instead of solely relying on $y$ to inject the semantics, we explicitly reparameterize the source (or target) distribution using condition $y$. This can make the resulting flow trajectory simpler because the source distribution depends on the condition $y$—transporting mass becomes easier to learn.
>
> We hope this clarifies our intuition.
>
> ----
>
> - $\textbf{Reviewer comment (Q3)}$:  Performance difference between target-only and source-only CAR?
>
> Note that neither source-only nor target-only CAR consistently outperforms the other in our experiments. While both variants reliably improve upon the unconditional baseline, we find their relative advantage is small and depends on the dataset and hyper-parameters:
> 1) On 1-D synthetic data (Sec. 4.1), source-only CAR yields a very slightly lower Wasserstein distance than target-only (Fig. 3(a)).
> 2) On ImageNet, Table 3 shows target-only CAR delivers a small FID improvement over source-only. However, in our learning-rate ablation (Fig. 6, Appendix), neither variant shows a statistically significant edge.
>
>
> ----
>
> - $\textbf{Reviewer comment (Q4)}$: Meaning of cases (iii) and (iv)?
>
> Cases (iii) and (iv) in Claim 1 are theoretical degenerate "collapse" solutions that can be derived from Eq. (26) in the Appendix and require unbounded weights. When the scale of either the source or target distribution tends to infinity, it causes the counterpart distribution to appear infinitesimally narrow in comparison—effectively collapsing it to a point, leading to zero-cost loss. As noted in L164–165, such unbounded weights are not realistic in practice, so these solutions are unstable—any small perturbation causes the optimization to revert to cases (i) or (ii). We included them in Claim 1 for a theoretically comprehensive discussion and will add a remark in the revised manuscript to emphasize this.

---

> > ### Comment · Reviewer_naoP · 2025-08-07
> >
> > The issues I have raised have been mostly addressed, and I have updated my score accordingly.

---

### Decision · Program_Chairs · 2025-09-17

**Decision:**

Accept (spotlight)

**Comment:**

(a) This paper introduces Condition-Aware Reparameterization (CAR), a method to improve conditional generative modeling in flow-matching and diffusion frameworks. CAR introduces condition-dependent shifts of the source and/or target distributions, thus reducing trajectory lengths. The authors provide a theoretical analysis that identifies trivial zero-cost solutions under general reparameterizations and show that restricting to shift-only avoids these degeneracies. Experiments on synthetic data and ImageNet-256 demonstrate that CAR improves sample quality (e.g., they reduce FID from 2.07 to 1.68) with minimal computational overhead, while also shortening generative paths.

(b) The proposed method is clear and simple and the paper is well-written. CAR is easy to implement, adds minimal overhead, and integrates naturally into existing conditional flow-matching setups. The authors provide theoretical insights, demonstrating that allowing certain families of reparameterizations can lead to degenerate, trivial solutions. They show that, by restricting reparameterizations to shift-only transformations, CAR avoids these degenerate cases. This work has practical impact. The authors demonstrate experimentally that the method achieves strong improvements on a competitive ImageNet benchmark.

(c) The reviewers raised several concerns. One limitation is in the scope of the experiments. Experiments are mainly on ImageNet-256, and performance on other datasets and modalities (eg, text and audio) remains unexplored. The presentation of the theoretical analysis needs more work. Some parts of the analysis (eg, degenerate solutions, proportional collapse, independence assumptions) were found confusing or underspecified. The reviewers pointed out missing ablations and comparisons.

(d) The method is simple, broadly applicable, and well-motivated both theoretically and empirically. Despite the issues on incomplete experiments and the unclear theory presentation, the proposed method is found valuable. The performance improvement on ImageNet-256 is significant. Reviewers consistently valued the clarity, theoretical framing, and practical relevance, with two recommending full acceptance and the others borderline accept. The main limitations leave room for future work but do not undermine the contribution. The paper is a candidate for spotlight, as it presents an approach that would be of interest to a broad audience.

(e) The main points raised concerned: 1) the scope of experiments (only ImageNet and toy data), 2) the clarity of theoretical results on degenerate solutions, 3) whether unconditional generation could benefit, and 4) whether more general transformations could be explored. The authors provided clarifications during rebuttal, which reviewers felt addressed most of the concerns. Several reviewers raised their scores (e.g., naoP updated significance and score upward, rueW upgraded to accept, n8K3 confirmed rebuttal addressed concerns). The consensus converged to acceptance, with reviewers acknowledging CAR as a simple and effective addition to conditional flow matching despite limited experimental breadth.